# LinguaMap: Which Layers of LLMs Speak Your Language and How to Tune Them?

**J. Ben Tamo[1]\*, Daniel Carlander-Reuterfelt [2], Jonathan Rubin[2], Oleg Poliannikov[2],
Dezhi Hong[2], Mingxian Wang[2]**

**[1]Georgia Institute of Technology**          **[2]Amazon**

## Abstract

Despite multilingual pretraining, large language models often struggle with non-English tasks, particularly in language control–the ability to respond in the intended language. We identify and characterize two key failure modes: the *multilingual transfer bottleneck* (correct language, incorrect task response) and the *language consistency bottleneck* (correct task response, wrong language). To systematically surface these issues, we design a four-scenario evaluation protocol spanning MMLU, MGSM, and XQuAD benchmarks. To probe these issues with interpretability, we extend logit lens analysis to track language probabilities layer by layer and compute cross-lingual semantic similarity of hidden states. The results reveal a three-phase internal structure: early layers align inputs into shared semantic space, middle layers perform task reasoning, and late layers drive language-specific generation. Guided by these insights, we introduce *selective fine-tuning* of only the final layers responsible for language control. On Qwen-3-32B and Bloom-7.1B, this method achieves over $98\%$ language consistency across six languages while fine-tuning only 3–5% of parameters, without sacrificing task accuracy. Importantly, this result is nearly identical to that of full-scope fine-tuning (e.g., $> 98\%$ language consistency for both methods across all prompt scenarios) but uses a fraction of the computational resources. To the best of our knowledge, this is the first approach to leverage *layer-localization of language control* for efficient multilingual adaptation.

## 1 Introduction

The growing deployment of multilingual large language models (mLLMs) promises to bridge linguistic divides and democratize access to information across the world's languages. Early models such as mBERT (Devlin et al., 2019), XLM-R (Conneau et al., 2020), and mT5 (Xue et al., 2020) demonstrated impressive cross-lingual generalization, while more recent large-scale LLMs, such as PaLM-2 (Anil et al., 2023) and GPT-4, have shown even stronger multilingual capabilities, often without explicit multilingual supervision. Alongside these proprietary models, an expanding ecosystem of openly available multilingual LLMs has emerged, including BLOOM (Le Scao et al., 2022), LLaMA (Touvron et al., 2023), and Qwen (Yang et al., 2025). Despite this progress, we find that these models still exhibit persistent failures in language control, namely, the ability to respond in the intended language, even when they correctly solve the underlying task.

To systematically characterize multilingual failures, we introduce a targeted evaluation framework with four zero-shot prompt variants, each isolating a different aspect of language control. (1) Monolingual Direct Prompting tests whether models can follow instructions and respond exclusively in the target language; (2) Code-Switched Prompting examines robustness to mixed-language input; (3) Bilingual Answer Prompting probes language preference when correct answers are presented in both the target language and English; and (4) English Distractor Prompting tests resistance to incorrect English alternatives.

---

\*Work completed while at Amazon.
Correspondence to `jtamo3@gatech.edu`, {`poliann, mingxiw`}`@amazon.com`

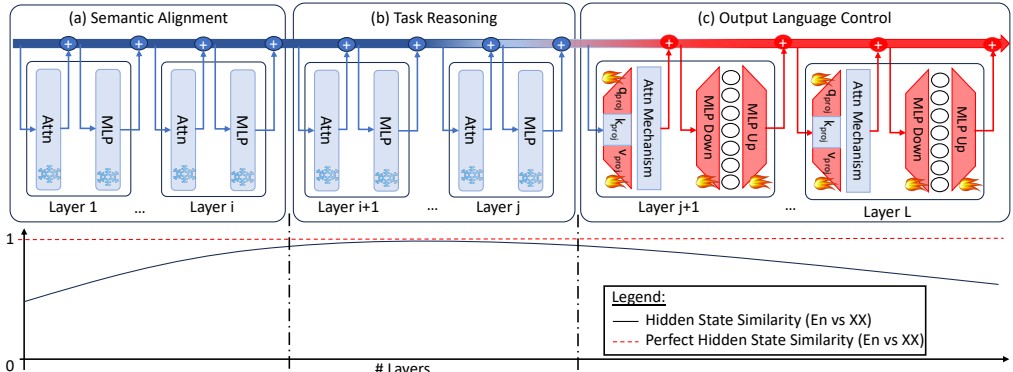

Figure 1: Overview of Selective Finetuning for Language Control: Early layers are frozen to preserve semantic alignment, mid layers maintain task reasoning, and only upper layers are finetuned to introduce language-specific output control, enabling efficient multilingual adaptation with minimal disruption to core model capabilities.

This evaluation reveals two failure modes: (1) language consistency bottleneck, where a model generates the correct answer but in the wrong language; (2) multilingual transfer bottleneck, where a model generates output in the correct language but fails tasks it can solve in English. These failures highlight a deeper disconnect between task competence and language control in mLLMs, suggesting that they are governed by distinct internal mechanisms.

Failures in language control often stem from Anglocentric pretraining, limitations in shared multilingual representations, and interference across typologically diverse languages (Huang et al., 2024; Zhao et al., 2024; Papadimitriou et al., 2023). While prior work has explored fine-tuning (Artetxe et al., 2020b), language-specific embeddings (Cao et al., 2020), prompt engineering (Shi et al., 2023; Vatsal et al., 2025), and monolingual specialization (Dobler & de Melo, 2023), these approaches often face trade-offs in scalability and coverage. Understanding how internal representations shape cross-lingual behavior remains an open challenge. We ask: ***Where in the model do language-specific behaviors–such as language consistency, dominance bias, and multilingual interference–reside, and can they be isolated to enable efficient and effective multilingual adaptation?***

We answer this by taking a structural, mechanistic view. We apply logit lens analysis of language token probabilities and semantic similarity evaluation of multilingual hidden states. Both analyses converge on a three-space structure, a semantic alignment phase, a reasoning phase, and a language output phase, previously hypothesized in recent studies (Zhao et al., 2024; Wendler et al., 2024; Etxaniz et al., 2024; Schut et al., 2025; Lindsey et al., 2025): (i) Early layers gradually normalize language inputs into a shared semantic space, (ii) mid layers perform task reasoning, and (iii) late layers control language-specific output.

Building on this understanding, we introduce layer-wise selective fine-tuning, a lightweight method that targets only the final output layers responsible for language control. Applied to models like Qwen-3-32B and Bloom-7.1B, it improves language consistency from <20% to 98+% across six languages, while preserving task performance and updating fewer parameters than full fine-tuning.

**Main contributions**. This work (1) introduces a framework for evaluating language control in mLLMs, incorporating systematic prompt variation to diagnose multilingual failure modes, (2) uncovers and validates a three-space structure in mLLMs, where distinct layers specialize in semantic alignment, reasoning, and language generation, and (3) proposes and validates layer-wise selective fine-tuning as an efficient and effective method to correct language consistency failures without compromising performance.

## 2 RELATED WORK

Recent advances in multilingual language models have revealed deep structural asymmetries favoring English, even in models trained across diverse linguistic corpora.

## 2.1 LATENT ENGLISH DOMINANCE IN mLLMs

Multiple studies reveal that state-of-the-art multilingual transformers often process non-English inputs via internal English representations. Schut et al. (2025) and Wendler et al. (2024) empirically confirm models like LLaMA-2 implicitly reason in intermediate English-based latent spaces, even when inputs and outputs are in other languages. Wendler et al. (2024) formalize this through their multilingual workflow hypothesis, showing that multilingual reasoning commonly pivots through English representations in intermediate layers. Complementing these findings, Lindsey et al. (2025) introduces the concept of multilingual circuits and further highlights that multilingual models often use English as the default internal representation, implying asymmetrical semantic spaces biased toward English. These findings collectively indicate an internal "English-thinking" phenomenon, contrasting the apparent multilingual capabilities observed externally.

## 2.2 LANGUAGE LOCALIZATION AND NEURON INSIGHTS

Building interpretability research, Tang et al. (2024) show that multilingual models contain distinct clusters of neurons selectively responsive to particular languages. Wang et al. (2024) further confirm that input/output layers exhibit stronger language-specific activation, whereas middle layers encode language-agnostic concepts. Zhao et al. (2024) confirm this belief through parallel language-neuron detection and conclude that there are three spaces: input, conceptual, and output spaces. These insights sparked research in language adaptability. Pfeiffer et al. (2020) introduce an invertible adapter architecture for adapting a pre-trained multilingual model to a new language. Huo et al. (2025) propose deep supervision fine-tuning, explicitly aligning internal representations across layers, significantly reducing latent English bias. Similarly, Liu & Niehues (2025) emphasize explicit representational alignment during fine-tuning at intermediate layers, promoting cross-lingual semantic consistency and improving zero-shot transfer. Kew et al. (2024) explores how much multilingual finetuning is needed to turn English-centric models into "polyglots," and Zhong et al. (2025) investigates which internal language representations non-English-centric models use during inference.

Collectively, these works suggest that multilingualism in LLMs is not uniformly supported at all representational layers. While embedding layers may provide aligned token representations across languages, deeper layers exhibit emergent specialization or drift toward dominant languages. Building upon these insights, our work further identifies language-specific processing layers and, based on this finding, proposes efficient fine-tuning strategies to enhance multilingual performance.

## 3 A PROMPT-BASED FRAMEWORK FOR DIAGNOSING LANGUAGE CONTROL

### 3.1 PROMPT STRUCTURE AND COMPONENT DESIGN

Our framework consists of four zero-shot prompting variants, each probing a distinct aspect of language control. We define a prompt as comprising three main input components: **Preamble (P)**—the metadata that frames the task; **Instruction (I)**—the explicit directive describing the task to perform; and **Question (Q)**—the task content itself, such as a question or passage. The mLLMs respond with two output components: **Reasoning (R)** and **Answer (A)**.

We evaluate performance across three multilingual benchmarks: MMLU Hendrycks et al. (2021), MGSM Shi et al. (2023), and XQuAD Artetxe et al. (2020a), which cover multiple-choice (MMLU), generative reasoning (MGSM), and extractive span-based answering (XQuAD). The zero-shot prompt variants (see Figure 6 in Appendix) include:

- Monolingual Direct Prompting, which tests baseline fidelity when both instructions and content are in the target language;
- Code-Switched Prompting, where the instruction or metadata is in one language (e.g., English), while the task content (e.g., the question) is written in the target language, testing the model's ability to resolve linguistic context under mixed-language input;
- English Distractor Prompting, which includes incorrect English answers to test rejection of misleading output.
- Bilingual Answer Prompting, which presents correct answers in both the target language and English to probe language preference;

To isolate linguistic effects, we keep the semantic content for the correct answer constant across languages and measure language consistency, whether responses are in the intended language, regardless of correctness. We apply each prompt variant to a standardized set of questions across six typologically and scripturally diverse languages: English, French, Spanish, Arabic, Hindi, and Japanese.

## 3.2 MULTILINGUAL BENCHMARK SETUP AND METRICS

To analyze multilingual model behavior more precisely, we decompose performance along two orthogonal axes: task accuracy and language consistency. Task accuracy evaluates whether the model provides the correct answer, regardless of the output language. Language consistency refers to whether the response is delivered entirely in the intended target language. We focus on the two most revealing failure modes:

- **Multilingual Transfer Bottleneck**: The model responds in the correct language but fails to provide the correct answer, despite likely being capable of solving the task in another language (e.g., English).
- **Language Consistency Bottleneck**: The model produces a correct answer but in the wrong language, indicating difficulty in adhering to the requested linguistic context.

Language consistency is computed as the proportion of responses whose primary language matches the target language. Let $y_i$ be the model output for example $i$, and $\text{Lang}_{\text{target}}$ be the expected output language. Let $\text{Lang}(y_i)$ be the predicted language of the model output, determined using the LangDetect language identifier by Shuyo (2010).

## 3.3 FINDINGS: WHEN AND HOW MLLMS FAIL

We evaluate two multilingual LLMs, Qwen-3-32B, and BLOOM-7.1B, on MMLU, XQuAD, and MGSM under zero-shot settings, focusing on two core dimensions: task performance and language consistency, as presented in Table 1. These benchmarks collectively span factual knowledge, multilingual reasoning, and mathematical problem solving across diverse languages.

Table 1 reports average scores across all evaluated languages for each dataset, MMLU (6: en, es, fr, ar, hi, ja), MGSM (3: en, fr, ja), and XQuAD (4: en, es, ar, hi). Switching from monolingual to codeswitched prompts often leaves average task accuracy largely intact but can sharply degrade language consistency. For example, Qwen-3-32B maintains strong average MMLU accuracy (60.5% under code-switched prompts vs. 51.77% monolingual), yet its average language consistency drops from 45.17% to just 8.35%. BLOOM-7.1B shows the same pattern; the language consistency across all three datasets drops while the task performance remains comparable or slightly better.

When averaged across languages and prompt types, Qwen-3-32B consistently achieves the highest task scores (e.g., 66.6% MGSM monolingual, 55.54 F1 on XQuAD monolingual) but suffers severe language consistency losses, often into single digits, whenever prompts mix languages or include English distractors. BLOOM-7.1B, in contrast, underperforms on both metrics, with average MGSM accuracies ≤0.67% and XQuAD F1 scores frequently under 7%, despite occasionally high consistency in certain monolingual conditions. These trends suggest that Qwen-3-32B is optimized for multilingual task utility, and BLOOM, despite moderate language control, fails to engage with task semantics.

Stress tests expose finer-grained weaknesses. Under English-distractor prompting (MMLU; averaged across languages), Qwen-3-32B's language consistency drops from 45.17% to 23.54%, while accuracy declines from 51.77% to 36.93%. Across prompt types, damage often correlates with language distance in per-language breakdowns (Appendix Tables 4, 5, 6), suggesting that shared subword inventory may cushion losses. BLOOM's scores remain uniformly low, with average MGSM accuracies ≤0.67% and XQuAD F1 scores frequently under 7%, reinforcing that its limitations are capacity-driven rather than prompt-specific.

These results reflect differing model priorities: some architectures, like Qwen, favor task success even at the cost of language control, while others, like BLOOM, attempt to enforce language control more strictly. Qwen-3-32B consistently achieves high accuracy across domains and languages but

Table 1: Multilingual Trade-offs Across Prompting Strategies: Evaluated MMLU (6 languages: en, es, fr, ar, hi, ja), MGSM (3: en, fr, ja), and XQuAD (4: en, es, ar, hi), Qwen-3-32B achieves the highest task performance but suffers major drops in language consistency; Bloom-7.1B lags on both. Overall, robustness to cross-lingual prompt perturbations often comes at the expense of peak task accuracy. Complete breakdowns for all datasets are provided in Appendix Tables 4, 5, 6.

| Prompting | Dataset | Bloom 7.1B | | Qwen-3 32B | |
|---|---|---|---|---|---|
| | | Language Consistency (%) | Task Performance (%) | Language Consistency (%) | Task Performance (%) |
| **Monolingual** **P, I, Q - (X)** | MMLU | 67.98 | 15.83 | 45.17 | 51.77 |
| | MGSM | 34.00 | 0.67 | 65.56 | 66.60 |
| | XQuAD | 98.32 | 4.18 | 81.05 | 55.54 |
| **Code-Switched** **P, I - (EN), Q(X)** | MMLU | 29.49 | 22.31 | 8.35 | 60.50 |
| | MGSM | 18.41 | 0.40 | 6.84 | 57.00 |
| | XQuAD | 71.23 | 6.58 | 11.01 | 52.65 |
| **English-Distractor** **I - (X), Q(X & EN)** | MMLU | 40.00 | 10.51 | 23.54 | 36.93 |
| | XQuAD | 69.44 | 0.67 | 41.99 | 15.81 |
| **Bilingual-Answer** **I - (X), Q(X & EN)** | MMLU | 59.61 | 9.36 | 23.50 | 35.76 |

struggles with stable language control, particularly under mixed prompt languages. BLOOM-7.1B, despite its fluent output, lacks the semantic depth required for effective multilingual reasoning.

# 4 WHERE LANGUAGE CONTROL EMERGES: LAYER-WISE INTERPRETABILITY

Prompt-level behavior shows failures in language control, particularly under code-switching, but the mechanisms driving output language choice in multilingual LLMs remain unclear. We use interpretability tools to probe internal activations, identifying where language control emerges and how inconsistencies are encoded in hidden representations.

## 4.1 METHODS FOR PROBING INTERNAL REPRESENTATION

### 4.1.1 DECODING INTERNAL LANGUAGE PROBABILITIES WITH THE LOGIT LENS

To trace the evolution of language preferences in multilingual LLMs, we use logit lens decoding (nostalgebraist et al., 2021), which projects intermediate hidden states onto the output vocabulary via the model's language modeling head. At each layer $l$, we compute pseudo-logits by projecting the intermediate state $\mathbf{h}_i^{(l)}$ through the unembedding matrix $\mathbf{U} \in \mathbb{R}^{|V| \times d}$:

$$\mathbf{z}_{i,t}^{(l)} = \left[\mathbf{U}\,\mathbf{h}_i^{(l)}\right]_t = \mathbf{u}_t^\top\,\mathbf{h}_i^{(l)}, \tag{1}$$

where $\mathbf{u}_t \in \mathbb{R}^d$ is the embedding of vocabulary token $t$. These pseudo-logits approximate the model's next-token distribution at each layer.

For each position $i$ in the generation, we decode the most likely token from the pseudo-logits at every layer, yielding an $M$-length intermediate sequence per layer when the model generates $M$ tokens. After reconstructing full words from subword tokens, we compute language probabilities using the $langdetect$ language identifier library (Shuyo, 2010). Operating at the word level avoids the ambiguity introduced by multilingual subword overlap. The identifier compares each reconstructed word against pre-trained language profiles derived from character-distribution statistics and returns normalized probabilities over languages:

$$p_j^{(l)}(\ell) = \frac{\exp\big(s(\hat{y}_j^{(l)}, \ell)\big)}{\sum_{\ell' \in \mathcal{L}} \exp\big(s(\hat{y}_j^{(l)}, \ell')\big)}, \tag{2}$$

where $p_j^{(l)}(\ell)$ denotes the probability that decoded word $\hat{y}_j^{(l)}$ belongs to language $\ell$. This word-level approach ensures that language identification relies on words from full decoded sequences,

providing a more stable and robust signal than subword-level approaches in multilingual settings. By aggregating word-level language predictions, we estimate the language probability mass at each layer and track shifts in preference between the target language and dominant alternatives (typically English).

$$P^{(l)}(\ell) = \frac{1}{M} \sum_{j=1}^{M} p_j^{(l)}(\ell),\tag{3}$$

which represents the average probability mass assigned to language $\ell$ at layer $l$. Tracking $P^{(l)}(\ell)$ across layers yields the trajectory of language drift during generation.

### 4.1.2 HIDDEN STATE SIMILARITY ANALYSIS

We perform a layer-wise analysis of hidden state similarity across language pairs using cosine similarity. Given a set of aligned prompts $\{(x_n^{(E)}, x_n^{(A)})\}_{n=1}^{N}$, where each pair consists of semantically equivalent inputs in English and another language $A$ (e.g., Spanish), we pass each prompt through the model and extract hidden states at each layer $\ell \in \{0, \dots, L\}$, including the embedding layer. We compare the internal representations layer-by-layer to determine where they begin to diverge.

For each input, the hidden states at layer $\ell$ are denoted $h_\ell^{(E,n)} \in \mathbb{R}^{T_n^{(E)} \times d}$ and $h_\ell^{(A,n)} \in \mathbb{R}^{T_n^{(A)} \times d}$, where $d$ is the hidden size and $T$ is the sequence length. To obtain a fixed-size prompt representation per layer, we apply mean pooling over all token embeddings in the input sequence:

$$\bar{h}_\ell^{(E,n)} = \frac{1}{T_n^{(E)}} \sum_{t=1}^{T_n^{(E)}} h_{\ell,t}^{(E,n)}, \quad \bar{h}_\ell^{(A,n)} = \frac{1}{T_n^{(A)}} \sum_{t=1}^{T_n^{(A)}} h_{\ell,t}^{(A,n)}.\tag{4}$$

We then compute the cosine similarity between the mean-pooled representations:

$$s_\ell^{(n)} = \frac{\langle \bar{h}_\ell^{(E,n)}, \bar{h}_\ell^{(A,n)} \rangle}{\|\bar{h}_\ell^{(E,n)}\| \cdot \|\bar{h}_\ell^{(A,n)}\|}.\tag{5}$$

Aggregating across the dataset yields the average and standard deviation of similarity per layer:

$$\bar{S}_\ell = \frac{1}{N} \sum_{n=1}^{N} s_\ell^{(n)}, \quad \sigma_\ell = \sqrt{\frac{1}{N} \sum_{n=1}^{N} \left( s_\ell^{(n)} - \bar{S}_\ell \right)^2}.\tag{6}$$

This mean-pooled prompt similarity analysis offers a high-level but interpretable view of how representations evolve across layers. Mean-pooled hidden-state cosine similarity (equation 5 ad equation 6) robustly captures global, sequence-level semantic alignment, even when cross-lingual tokenization differs substantially across languages. Although this abstraction hides token-level divergence in attention or contextual span, token-wise comparisons are highly sensitive to tokenization mismatch and require non-trivial alignment across sequences of different lengths, often introducing noise that obscures the underlying conceptual structure.

### 4.2 FINDINGS ON LAYER LANGUAGE CONTROL AND REPRESENTATION

We apply these interpretability methods across monolingual and code-switched prompting, in five non-English languages (ES, FR, AR, HI, JA). English is never used as the sole prompt language. Our analysis compares how much target language control vs. English dominance emerges at different network depths. Figures 2 and 3 jointly trace how multilingual LLMs control and represent language across layers. The logit lens analysis shows how word-level language probabilities evolve, while hidden-state similarity analysis examines the degree to which parallel prompts in different languages occupy a shared representation space.

### 4.2.1 MODEL-SPECIFIC PATTERNS IN LANGUAGE CONTROL AND REPRESENTATION

In Bloom-7.1B, monolingual prompts yield high target-language probabilities for Arabic, Hindi, and Japanese (0.4–1 up to layer 10), with Spanish and French slightly lower but still exceeding English.

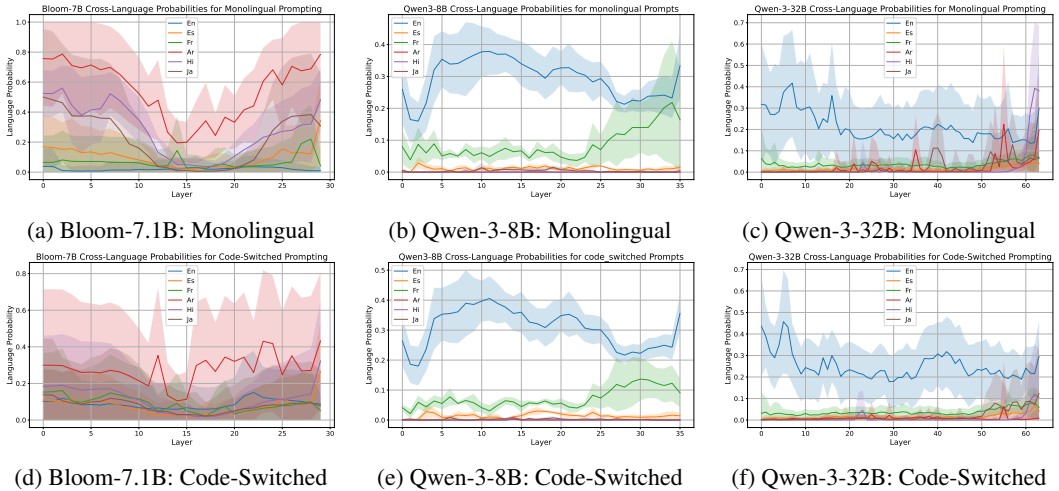

(a) Bloom-7.1B: Monolingual    (b) Qwen-3-8B: Monolingual    (c) Qwen-3-32B: Monolingual

(d) Bloom-7.1B: Code-Switched    (e) Qwen-3-8B: Code-Switched    (f) Qwen-3-32B: Code-Switched

Figure 2: Cross-Language Probability by Layer under Monolingual and Code-Switched Prompting on MMLU. In Qwen, early layers are relatively biased to English, middle layers sustain English bias, and final layers shift toward language-specific processing. However, code-switching disrupts this control, especially in Qwen. Bloom exhibits more language-specific layers with no bias.

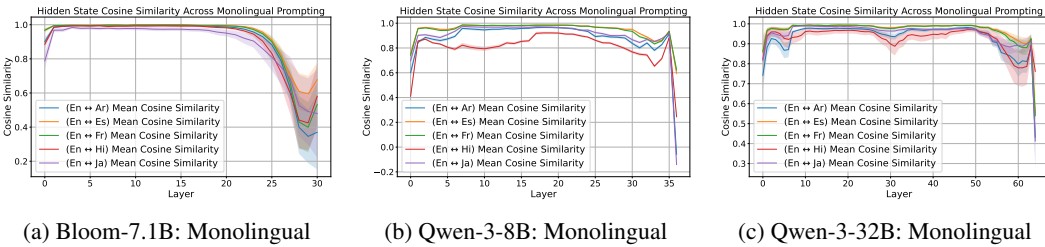

(a) Bloom-7.1B: Monolingual    (b) Qwen-3-8B: Monolingual    (c) Qwen-3-32B: Monolingual

Figure 3: Layer-wise hidden-state cosine similarity for monolingual MMLU prompts. Each sub-figure shows similarity between English and five target languages (ES, FR, JA, AR, HI) across the embedding output and transformer layers. Similarity rises sharply in early layers, remains stable in mid-layers where cross-lingual semantic alignment is strongest, and declines in the final layers.

These probabilities weaken in mid-layers but recover strongly after layer 20, while English remains consistently suppressed ($<$0.1). The wide shaded regions (standard deviations) reveal substantial variability across layers, suggesting that Bloom's intermediate layers mix cross-lingual features, producing ambiguous intermediate decodings and thus unstable language probability estimates. Under code-switching, however, target-language control collapses (probabilities $<$0.2), with only partial recovery for Arabic, Hindi, and Spanish. Representation-wise, Bloom shows a rapid increase to high cross-lingual similarity in early layers. Middle layers maintain strong similarity (0.97–0.99), implying a strongly language-invariant semantic space. Only in late layers (24–30) do sharp divergences appear, especially for En–Ar (to 0.36) and En–Hi (to 0.42), reflecting linguistic divergence: language control emerges. Qwen-3-8B exhibits a consistently English-dominant: its monolingual prompting behavior (Figure 2) shows English dominating generation bias across most layers, while target-language probabilities (ES, FR, AR, HI, JA) start weak, remain suppressed through the middle layers, and only FR recovers partially after layer 25. Similarly, under code-switched prompting (Figure 2), Qwen-3-8B's language control collapses fully: target-language probabilities remain suppressed in the final layers, and English becomes the sole generation language.

In Qwen-3-32B, English dominates early regardless of input language, with target-language probabilities rising only after layer 55. This layer marks the data-driven onset of language-specific generation, defined operationally by the convergence of two independent and empirically easy-to-identify indicators: the layer where the target language probability first surpasses English, and the layer where cross-lingual hidden-state similarity sustains a divergence from the stable, middle-layer align-

ment. Even after this emergence, recovery is incomplete, and under code-switching, re-grounding fails altogether. Hidden-state analysis shows very high similarity across languages in middle layers 6–55 (En–Es/Fr near 0.99, En–Ar/Ja 0.95–0.97, En–Hi just under 0.9). After layer 55, similarity diverges slightly, which does not translate into effective language control: English remains entrenched as the dominant generation bias.

### 4.2.2 Cross-Linguistic Differences in Representation and Control

Across models, we observe a three-phase structure, early convergence to a shared semantic space, stable middle layers, and late divergence into language-specific generation, but the stability of language control differs. Bloom-7.1B shows high early target-language probabilities but with large variance across layers, reflecting ambiguous intermediate decodings where hidden states straddle multiple languages. In contrast, Qwen-3-32B is stable but strongly English-biased: English dominates early and mid layers, target-language probabilities rise only after layer 55, and recovery fails under code-switching. These contrasts suggest that instability in Bloom arises from ambiguous intermediate representations, while Qwen's consistency reflects entrenched bias.

## 5 Layer-wise Selective Fine-Tuning for Language Consistency

The analyses in Sections 3 and 4 demonstrate that mLLMs often lose language control under adversarial prompts, a failure linked to unstable late-layer re-grounding. To address this, we propose layer-wise selective supervised fine-tuning (SFT) that targets language control mechanisms without full model retraining.

### 5.1 How to Tune Language Control: Selective Supervised Fine-Tuning

Our goal is to reinforce language consistency, the model's ability to produce outputs strictly in the intended language, while minimizing interference with general task competence coverage.

Consider a pretrained model with parameter set $\theta = \{\theta_1, \theta_2, \ldots, \theta_L, \theta_{\text{head}}\}$ where $\theta_\ell$ corresponds to layer $\ell$, and $\theta_{\text{head}}$ is the embedding and LM head. Rather than tuning all layers, we update only a subset $\mathcal{S} \subseteq \{1, \ldots, L\}$, typically the last $k$ layers, where language-specific generation behavior emerges.

We define selective SFT as fine-tuning only a subset of parameters $\theta_{\mathcal{S}}$, while keeping the remaining parameters $\theta_{-\mathcal{S}}$ frozen. Given training data $\{(x_i, y_i)\}_{n=1}^N$, the optimization objective is to minimize:

$$\mathcal{L}_{\text{Selective-SFT}}(\theta_{\mathcal{S}}) = -\sum_{i=1}^N \log P(y_i \mid x_i; \theta_{\mathcal{S}}), \tag{7}$$

where gradients are computed only with respect to $\theta_{\mathcal{S}}$. This formulation isolates adaptation to the selected components while leveraging the frozen parameters to preserve the pretrained model's semantic alignment and reasoning capacity.

To evaluate Selective SFT, we fine-tuned on a domain-focused MMLU subset covering five business subjects (ethics, marketing, management, accounting, public relations) across five languages (Spanish, French, Arabic, Hindi, Japanese). From the pool of correctly answered examples (verified with Claude 3.5 Sonnet), we sampled 500 per subject, yielding 2,500 examples split 80/20 into training and validation sets. Each instance was augmented with chain-of-thought reasoning traces aligned with the question's language. Prompts followed a five-part template (Preamble (P), Instruction (I), Question (Q), Reasoning (R), Answer (A)), with loss restricted to the Q, R, A tokens while P and I remained frozen context:

$$\mathcal{L}_{\text{Selective-SFT}}^{\text{masked}}(\theta_{\mathcal{S}}) = -\sum_{i=1}^N m_i \cdot \log P(y_i \mid Q_i, R_i, A_i; \theta_{\mathcal{S}}), \tag{8}$$

where $m_i \in \{0, 1\}$ masks tokens outside the Q, R, A, and gradients are applied only to $\theta_{\mathcal{S}}$.

An ablation varying tuned last layers $(1, \ldots, n)$ and epochs (1–5) (see Appendix Tables 10, 11) showed that the optimal configuration was the last layer at 5 epochs for Bloom-7.1B and the last two layers at 5 epochs for Qwen-3-32B.

Table 2: Impact of fine-tuning on language consistency and task performance for Qwen-3-32B and Bloom-7.1B on MGSM, MMLU, and XQuAD. Both models were fine-tuned with code-switched prompts in the MMLU Business domain across six languages, then evaluated on MMLU non-Business subjects (52 in total), MGSM, and XQuAD. Values represent averages across all evaluation languages for each dataset; full per-dataset results appear in the Appendix Tables 7, 8, 9.

| Prompting | Datasets | Model | Pre-Finetuning | | Full scope SFT | | Random Selective SFT | | Selective SFT | |
|---|---|---|---|---|---|---|---|---|---|---|
| | | | Language Cons. (%) | Task (%) | Language Cons. (%) | Task (%) | Language Cons. (%) | Task (%) | Language Cons. (%) | Task (%) |
| **# Trainable Param** | | Qwen-3-32B | NA | | 32B | | 1.5B | | 1.5B | |
| | | Bloom-7.1B | NA | | 7.1B | | 0.5B | | 0.5B | |
| **Monolingual P, I, Q - (X)** | **MGSM (Avg)** | Qwen-3-32B | 65.56 | 66.60 | **99.47** | **90.53** | 65.87 | 0.13 | 99.20 | 86.80 |
| | | Bloom-7.1B | 34.00 | 0.67 | **100** | 1.47 | 69.47 | 0.00 | **100.00** | **3.60** |
| | **XQuAD (Avg)** | Qwen-3-32B | 81.05 | 55.54 | **100.00** | **57.60** | 47.44 | 0.42 | 99.83 | 55.86 |
| | | Bloom-7.1B | 98.32 | 4.18 | **99.91** | 16.85 | 54.10 | 0.00 | 99.85 | **20.83** |
| **Code Switched P, I - (EN), Q(X)** | **MMLU (Avg)** | Qwen-3-32B | 8.35 | 60.51 | **99.87** | **78.84** | 98.30 | 1.67 | 99.62 | 74.44 |
| | | Bloom-7.1B | 29.49 | 22.31 | **99.87** | **33.72** | 55.56 | 0.00 | 98.66 | 21.14 |
| | **MGSM (Avg)** | Qwen-3-32B | 6.80 | 57.00 | 95.00 | **87.00** | 53.80 | 0.00 | **98.60** | 84.60 |
| | | Bloom-7.1B | 18.40 | 0.40 | **100** | 2.20 | 68.00 | 0.00 | 99.60 | 2.00 |
| | **XQuAD (Avg)** | Qwen-3-32B | 11.01 | 52.65 | **100.00** | 51.87 | 97.93 | 1.10 | **100.00** | **53.53** |
| | | Bloom-7.1B | 71.23 | 6.58 | **99.89** | 21.03 | 43.11 | 0.00 | 99.80 | **21.02** |
| **English Distractor I - (X), Q(X & EN)** | **XQuAD (Avg)** | Qwen-3-32B | 41.99 | 15.81 | 75.99 | 17.77 | 37.78 | 0.00 | **97.62** | **18.05** |
| | | Bloom-7.1B | 69.44 | 0.67 | 97.03 | **6.96** | 26.16 | 0.00 | **98.23** | **6.95** |

## 5.2 RESULTS AND FINDINGS

Table 2 indicates that, before fine-tuning, Qwen-3-32B shows moderate task accuracy but poor language control, achieving 66.6% on MGSM and 55.5% on XQuAD for monolingual prompts, while collapsing under code-switching with only 6–11% language consistency. Bloom-7.1B maintains higher consistency (34–98%) but is far weaker in task accuracy (0.4–15.8%), often producing text in the target language without solving the task. Full-scope SFT substantially improves Qwen, raising consistency to nearly 100% across all regimes and boosting task accuracy (e.g., MGSM from 66.6% to 90.5%). Code-switched settings, initially unstable, are restored above 95% language consistency with 78–87% task accuracy. Bloom also reaches near-perfect language consistency after full-scope SFT, though without comparable reasoning gains. Overall, full-scope SFT enforces consistent language use in both models, with Qwen uniquely leveraging this for improved task performance.

When tuning on a random subset of layers, both models collapse in task performance, despite retaining some language consistency. For example, Qwen's monolingual MGSM language consistency falls from nearly 100% (selective-sft) to 65.9% under random selective SFT, and Bloom shows a similar decline (from 100% to 69%). In contrast, Selective SFT recovers near-perfect language consistency across datasets. Both Qwen and Bloom maintain 99% language consistency in monolingual, code-switched, and English-distractor prompts. These results demonstrate that targeted layer adaptation preserves language consistency, whereas random selection destabilizes generation and erodes cross-lingual consistency. Selective SFT achieves nearly the same performance as full-scope SFT for both Qwen and Bloom, while requiring updates to only 3–5% of the parameters, compared to full-scope SFT. Random selective SFT is catastrophic, reinforcing the importance of principled parameter selection. Under English distractor prompts, selective SFT substantially improves language consistency, yet task performance remains weak, highlighting the need for more explicit reasoning-level disambiguation strategies in future work.

A critical diagnostic concern for our Selective SFT approach is whether the language control adjustments propagate backward, altering the semantic and reasoning alignments in the frozen middle layers. To address this, we conducted a full post-fine-tuning analysis using our original interpretability tools. As shown in Figure 4, the substantial increase in target-language probability is confined strictly to the late layers (the tuned region), confirming the successful localization of the intervention. Furthermore, Figure 5 provides direct empirical evidence of invariance: the high cross-lingual alignment signature in the semantically-aligned middle layers is fully preserved, remaining virtually identical to the pre-fine-tuning state. This analysis validates that Selective SFT successfully isolates the language control mechanism in the final layers without compromising the integrity of the model's core, language-invariant reasoning capabilities.

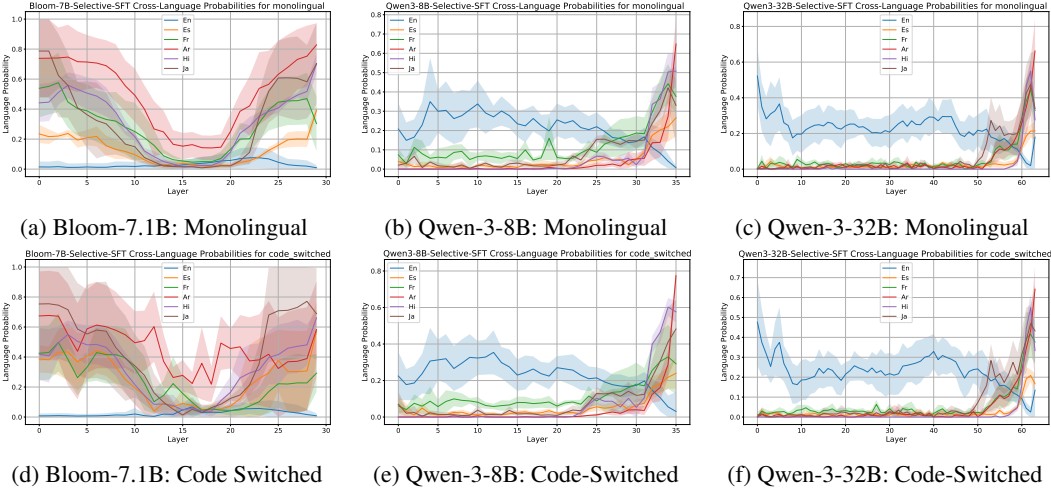

(a) Bloom-7.1B: Monolingual     (b) Qwen-3-8B: Monolingual     (c) Qwen-3-32B: Monolingual

(d) Bloom-7.1B: Code Switched     (e) Qwen-3-8B: Code-Switched     (f) Qwen-3-32B: Code-Switched

Figure 4: Post-Selective SFT Layer-wise Language Probability Trajectories. The plots, shown under Monolingual and Code-Switched prompting, confirm the localization of the intervention: non-English target-language probabilities substantially increase and dominate only in the late layers (the tuned region), with minimal change observed in the early and middle layers.

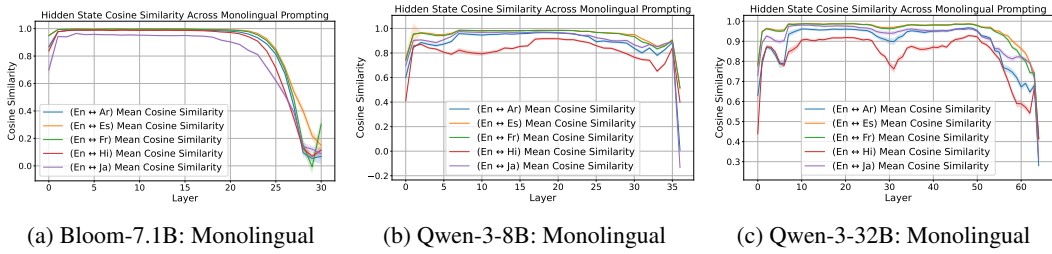

(a) Bloom-7.1B: Monolingual     (b) Qwen-3-8B: Monolingual     (c) Qwen-3-32B: Monolingual

Figure 5: Post-Selective SFT Hidden State Cosine Similarity Across Layers. The results demonstrate the stability of the frozen layers, maintaining the high cross-lingual similarity signature in the language-invariant middle layers and confirming that the language control adjustments did not propagate backward to alter the semantic alignment.

## 6 CONCLUSION

LinguaMap details how multilingual language control is distributed across layers in LLMs. By uncovering a robust three-phase structure, from shared semantic grounding to language-specific decoding, we pinpoint where models "think" versus where they "speak". This insight exposes distinct model tradeoffs: Qwen-3-32B excels at multilingual accurate task completion but often sacrifices language control; and Bloom-7.1B, while consistent in adhering to the intended language, struggles to reason reliably across languages. Guided by this structural lens, we introduce a selective fine-tuning strategy that focuses exclusively on the final layers responsible for language control. As LLMs continue scaling across cultures and scripts, LinguaMap offers both a diagnostic lens and a tool for aligning them with the world's linguistic diversity.

## 7 REPRODUCIBILITY STATEMENT

Below we summarize the key aspects of reproducibility, drawing on our study design and the supporting materials.

**Conceptual and Theoretical Transparency**

The paper provides a clear conceptual outline and prompt template used in multilingual stress tests, enabling readers to understand and replicate our approach. While the paper does not introduce

fundamentally new theory, it extends established theoretical tools with appropriate formal statements and proofs, and cites all relevant theoretical references.

**Dataset Usage**

Our experiments rely on publicly available datasets. We explain the motivation for choosing each dataset and provide proper citations to all external data sources. No new datasets are introduced, and all datasets used are already accessible to the research community, allowing others to replicate the experimental results without restrictions.

**Computational Experiments**

Table 3 specifies the number and range of hyperparameters explored during development and the criteria for selecting final parameter settings. We describe the computing infrastructure, including hardware specifications (CPU/GPU models, memory). Evaluation metrics are formally defined, and their selection is motivated.

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

## A  APPENDIX

### A.1  LLMs Usage

Large Language Models (LLMs) were used solely as general-purpose assistive tools to help polish the manuscript's language and to refine instructions within our multilingual prompt templates. Specifically, LLMs aided in improving grammar, clarity, and style, and in suggesting alternative phrasings for prompt templates. All scientific ideas, experimental design, and key arguments were conceived and written by the authors, and all factual statements were independently verified. Final prompt templates in English, French, Spanish, Arabic, Hindi, and Japanese were reviewed and validated by native speakers of each language to ensure accuracy and cultural appropriateness.

## A.2    Prompting Variants and Detailed Analysis by Dataset and Language

Despite impressive gains in cross-lingual generalization, multilingual LLMs often struggle with language control, the ability to produce responses in the intended language of the task. To systematically assess this underexplored failure mode, we use our targeted evaluation framework to isolate and stress-test different dimensions of language consistency across diverse multilingual settings (Figure 6).

Tables 4, 5 6 present per-language results for each dataset, revealing patterns that are averaged in Table 1. The fine-grained breakdown shows that the trade-off between reasoning ability and language consistency varies sharply by language, script, and prompt type.

For XQuAD (Table 4), Qwen-3-32B shows strong reasoning ability but is highly sensitive to prompt perturbations. Bloom-7.1B maintains moderate to high language consistency (>70%) across all languages and prompting styles, though its task performance remains limited (<12%), especially in non-English settings. In contrast, Qwen-3-32B exhibits strong task performance, but its language consistency varies significantly depending on the language and prompt type. Notably, Spanish shows the lowest language control across all prompting variants for Qwen-3-32B, suggesting a heightened susceptibility to interference; language consistency at 41.68% in the monolingual setting and collapses to near 1% (1.60%) in the code-switched variant. The presence of English leads to severe language collapse, particularly for Spanish and Hindi, despite relatively preserved task performance. Overall, while Bloom displays moderate to high language stability regardless of prompt structure, Qwen-3-32B's strong multilingual reasoning capabilities come with a trade-off in maintaining language control, especially when English is introduced.

In math tasks, Table 5 reveals that Bloom 7.1B consistently underperforms, showing both poor language control and very low task accuracy, particularly under multilingual conditions. In contrast, Qwen-3-32B exhibits a clear trade-off between language consistency and performance: it achieves high task accuracy across all prompting styles and languages, even as language consistency drops drastically, especially under code-switched prompts. For instance, under English code-switched prompts with French questions, language consistency falls to just 7.6%, while task accuracy remains high at 59.6%.

In MMLU, Table 6 reinforces the language-task trade-off seen in Qwen-3-32B: it delivers strong task accuracy across most languages, but language consistency sharply degrades under multilingual or mixed-language prompting. The problem is especially acute for Spanish and French, where language consistency drops below 1% in code-switched and distractor settings, despite task accuracy remaining above 75%. This pattern suggests that Qwen-3 32B frequently defaults to English when handling closely related languages, prioritizing task accuracy over maintaining language consistency. This behavior is further supported by Figure 2, which reveals that across all layers, the model performs reasoning in representations that are closely aligned with English, regardless of the input language. In contrast, Bloom-7.1B shows stronger language control, particularly for distant languages like Hindi and Arabic, but at the cost of much lower task performance, particularly in non-English scenarios. These trends indicate that language similarity with English leads to higher interference and loss of control in multilingual models like Qwen-3-32B.

Overall, Qwen-3-32B delivers the strongest reasoning performance but is prone to severe language drift under mixed-language prompts. While Bloom 7.1B maintains moderate to high language control across languages and prompting formats, its task accuracy remains low, highlighting its limited multilingual reasoning capabilities. Qwen-3-32B often answers accurately even when it fails to preserve the intended output language. This suggests that Qwen-3-32B prioritizes internal alignment with English representations. The degradation is especially pronounced for languages typologically closer to English (like Spanish and French), which appear more prone to collapse under English interference.

## A.3    Fine-tuning and Inference Settings

We perform full scope and selective fine-tuning on specific layers of large pre-trained language models. In the training setup, all model parameters are initially frozen, ensuring only selected layers are updated during fine-tuning. The layers to be fine-tuned are chosen from the output space. We use the `AdamW` optimizer with a learning rate of $1e^{-5}$ and the `OneCycleLR` scheduler to adjust the

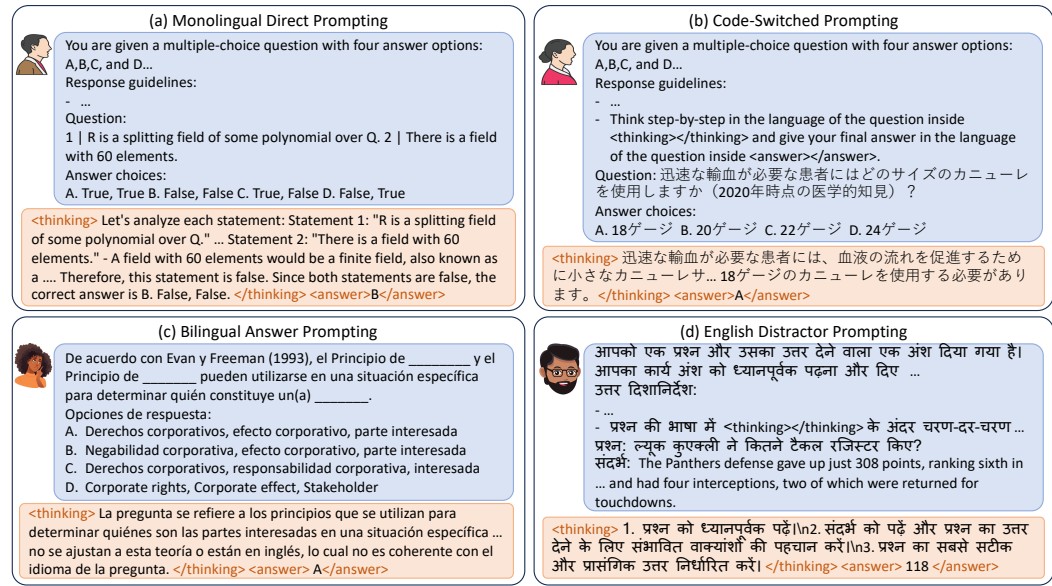

Figure 6: Overview of Multilingual Prompt Variants. Each variant isolates a specific aspect of multilingual generation: (a) Monolingual Direct Prompting tests baseline language adherence; (b) Code-Switched Prompting mixes instruction and task language to test robustness; (c) Bilingual Answer Prompting probes language preference by offering correct answers in both the target language and English; and (d) English Distractor Prompting tests resistance to dominant-language bias.

learning rate during training, starting from a small value and gradually increasing before decaying. After each batch, the loss is computed, and only the parameters in the selected layers are updated via backpropagation.

As part of the ablation study, we perform a grid search over two hyperparameters: the number of epochs (from 1 to 5) and the number of output space layers (from 1 to 5) fine-tuned. Table 10 . Table 11 shows that the best configuration for Qwen-3-32B is finetuning the last three layers. For all epochs, finetuning only the last three layers always achieves near-perfect language consistency 100% when tested on a subset (150) of non-business MMLU topics. The epoch number varies the task performance.

Table 3: Fine-tuning and Inference Parameters

| Hyperparameters | Values |
|---|---|
| Train Languages | ES, FR, AR, HI, JA |
| Train sample per Language | 500 |
| Train-Validation Split | 0.8/0.2 |
| Learning Rate | $1e^{-5}$ |
| Batch Size | 16 |
| Training Epochs | 1 to 5 |
| Number of Layers to Fine-Tune | 1 to 5 |
| Temperature | $1e^{-5}$ |
| Top k | 50 |
| Top p | 0.9 |
| Max New Tokens | 512 |
| Optimizer | AdamW |
| Learning Rate Scheduler | OneCycleLR |
| GPUs | 8x NVIDIA H100 |

Table 4: Language consistency and task performance (F1) on XQuAD across prompting conditions for Bloom-7B and Qwen-3-32B. Bloom maintains moderate to high language consistency (58 - 99%) but fails catastrophically in task performance (F1 <5% on average under monolingual prompting), revealing a disconnect between staying in-language and solving the task. In contrast, Qwen demonstrates stronger task ability but with uneven and unstable language control: high consistency in Arabic and English, but near-total collapse under Spanish and code-switched prompts.

| Prompting | Language | Bloom 7.1B | | Qwen-3 32B | |
|---|---|---|---|---|---|
| | | Language Consistency (%) | F1 Score (%) | Language Consistency (%) | F1 Score (%) |
| **Monolingual Direct** | P, I, Q - (EN) | 99.42 | 11.67 | 100 | 71.47 |
| | P, I, Q - (ES) | 98.24 | 3.50 | 41.68 | 56.17 |
| | P, I, Q - (AR) | 97.23 | 0.31 | 97.05 | 64.99 |
| | P, I, Q - (HI) | 98.40 | 1.24 | 85.46 | 29.51 |
| | **Average** | 98.32 | 4.18 | 81.05 | 55.54 |
| **Code Switched** | P, I -(EN), Q(ES) | 86.63 | 10.82 | 1.60 | 65.42 |
| | P, I -(EN), Q(AR) | 58.57 | 3.42 | 30.08 | 46.31 |
| | P, I -(EN), Q(HI) | 68.49 | 5.50 | 1.34 | 46.23 |
| | **Average** | 71.23 | 6.58 | 11.01 | 52.65 |
| **English Distractor** | I -(ES), Q(ES & EN) | 70.08 | 1.37 | 16.05 | 27.54 |
| | I -(AR), Q(AR & EN) | 69.41 | 0.36 | 62.27 | 9.81 |
| | I -(HI), Q(HI & EN) | 68.82 | 0.28 | 47.65 | 10.09 |
| | **Average** | 69.44 | 0.67 | 41.99 | 15.81 |

Table 5: Language consistency and task accuracy (%) on MGSM across different prompt variants for Bloom-7B and Qwen-3-32B. Bloom collapses on both axes, moderate to low language consistency, and near-zero task accuracy, indicating failure to maintain the target language and to solve the task. Qwen, by contrast, answers well, but its language control is brittle: perfect consistency in English and moderate in Japanese, yet severe collapse for French and under code-switching.

| Prompting | Language | Bloom 7.1B | | Qwen-3 32B | |
|---|---|---|---|---|---|
| | | Language Consistency (%) | Task Accuracy (%) | Language Consistency (%) | Task Accuracy (%) |
| **Monolingual Direct** | P, I, Q - (EN) | 61.20 | 1.20 | 100 | 66.00 |
| | P, I, Q - (FR) | 22.80 | 0.40 | 31.08 | 72.80 |
| | P, I, Q - (JA) | 18.00 | 0.40 | 65.60 | 60.99 |
| | **Average** | 34.00 | 0.67 | 65.56 | 66.60 |
| **Code Switched** | P, I -(EN), Q(FR) | 22.80 | 0.40 | 7.60 | 59.60 |
| | P, I -(EN), Q(JA) | 14.01 | 0.40 | 6.08 | 54.40 |
| | **Average** | 18.41 | 0.40 | 6.84 | 57.00 |

Table 6: Language consistency and task accuracy (%) on MMLU across different prompt variants for Bloom-7B and Qwen-3-32B. Bloom exhibits high language consistency in many settings but fails catastrophically at task accuracy, sometimes near zero, even when language consistency is high. Qwen flips the pattern: strong task performance in English, Spanish, and French ($\geq$70%), but fragile language control, collapsing almost completely in French, Spanish, and code-switched inputs. Under distractors and bilingual prompts, Bloom "sticks to the language but cannot answer," while Qwen "answers well but drifts in and out of the target language." The results expose a fundamental tension between being in-language and being correct in current multilingual LLMs.

| Prompting | Language | Bloom 7.1B | | Qwen-3 32B | |
|---|---|---|---|---|---|
| | | Language Consistency (%) | Task Accuracy (%) | Language Consistency (%) | Task Accuracy (%) |
| Monolingual Direct | P, I, Q - (EN) | 99.51 | 21.24 | 100 | 77.08 |
| | P, I, Q - (ES) | 50.22 | 16.59 | 0.31 | 76.33 |
| | P, I, Q - (AR) | 85.89 | 11.34 | 48.39 | 6.38 |
| | P, I, Q - (HI) | 96.79 | 10.76 | 69.09 | 34.23 |
| | P, I, Q - (FR) | 17.16 | 19.23 | 0.91 | 72.92 |
| | P, I, Q - (JA) | 58.33 | – | 52.32 | 43.68 |
| Code Switched | P, I -(EN), Q(ES) | 33.34 | 27.56 | 0.29 | 75.98 |
| | P, I -(EN), Q(AR) | 32.05 | 10.26 | 1.28 | 43.59 |
| | P, I -(EN), Q(HI) | 29.49 | 14.74 | 14.10 | 49.36 |
| | P, I -(EN), Q(FR) | 35.26 | 30.13 | 1.09 | 72.66 |
| | P, I -(EN), Q(JA) | 17.31 | 28.85 | 25.00 | 60.90 |
| English Distractor | I -(ES), Q(ES&EN) | 67.31 | 15.38 | 5.33 | 77.40 |
| | I -(AR), Q(AR&EN) | 37.82 | 6.41 | 18.58 | 4.05 |
| | I -(HI), Q(HI&EN) | 41.03 | 3.85 | 47.38 | 14.54 |
| | I -(FR), Q(FR&EN) | 25.64 | 26.92 | 0.24 | 75.84 |
| | I -(JA), Q(JA&EN) | 28.21 | 0.00 | 13.69 | 15.23 |
| Bilingual Answer | I -(ES), Q(ES&EN) | 44.87 | 19.87 | 15.96 | 55.69 |
| | I -(FR), Q(FR&EN) | 14.74 | 21.15 | 47.27 | 52.89 |
| | I -(AR), Q(AR&EN) | 85.26 | 0.00 | 12.77 | 6.31 |
| | I -(HI), Q(HI&EN) | 96.15 | 1.92 | 31.00 | 23.04 |
| | I -(JA), Q(JA&EN) | 57.05 | 3.85 | 10.50 | 40.85 |

Table 7: Performance comparison of Qwen-3-32B and Bloom-7.1B on the MMLU dataset. Models were trained using code-switched prompts in the Business domain across six languages and evaluated on a subset of non-Business domains.

| Language | Model | Pre-Finetuning | | Full scope SFT | | Selective SFT | |
|---|---|---|---|---|---|---|---|
| | | Language Cons. (%) | Acc. (%) | Language Cons. (%) | Acc. (%) | Language Cons. (%) | Acc. (%) |
| P, I -(EN), Q(ES) | Qwen-3-32B | 1.28 | 76.92 | 100 | 87.18 | 99.04 | 88.46 |
| | Bloom-7.1B | 33.34 | 27.56 | 99.36 | 35.90 | 98.08 | 26.92 |
| P, I -(EN), Q(FR) | Qwen-3-32B | 1.92 | 71.79 | 100 | 90.38 | 100 | 85.58 |
| | Bloom-7.1B | 35.26 | 30.13 | 100 | 35.90 | 98.08 | 23.08 |
| P, I -(EN), Q(HI) | Qwen-3-32B | 14.10 | 49.36 | 99.36 | 46.15 | 100 | 47.12 |
| | Bloom-7.1B | 29.49 | 14.74 | 100 | 33.97 | 98.08 | 20.19 |
| P, I -(EN), Q(AR) | Qwen-3-32B | 1.28 | 43.59 | 100 | 83.33 | 99.04 | 67.31 |
| | Bloom-7.1B | 32.05 | 10.26 | 100 | 32.05 | 100 | 17.31 |
| P, I -(EN), Q(JA) | Qwen-3-32B | 25.00 | 60.90 | 100 | 87.18 | 100 | 81.73 |
| | Bloom-7.1B | 17.31 | 28.85 | 100 | 30.77 | 99.04 | 18.27 |
| Code-Switched Average | Qwen-3-32B | 8.32 | 60.51 | 99.87 | 78.84 | 99.62 | 74.44 |
| | Bloom-7.1B | 29.49 | 22.31 | 99.87 | 33.72 | 98.66 | 21.14 |

Table 8: Comparison of Qwen-3-32B and Bloom-7.1B across monolingual and code-switched settings in English, French, and Japanese for pre-finetuning, full-scope SFT, random selective SFT, and targeted selective SFT. Qwen-3-32B shows strong gains from Selective SFT, especially in cross-lingual settings, while Bloom-7.1B remains fragile despite perfect language consistency post-finetuning, highlighting its limitations in multilingual task generalization. Selective SFT achieves near-parity with full-scope SFT in both consistency and accuracy, despite modifying fewer parameters.

| Language | Model | Pre-Finetuning | | Full scope SFT | | Random Selective SFT | | Selective SFT | |
|---|---|---|---|---|---|---|---|---|---|
| | | Language Cons. (%) | Acc. (%) | Language Cons. (%) | Acc. (%) | Language Cons. (%) | Acc. (%) | Language Cons. (%) | Acc. (%) |
| P, I, Q - (EN) | Qwen-3-32B | 100 | 66.00 | 98.80 | 97.20 | 1.20 | 0.40 | 99.60 | 95.60 |
| | Bloom-7.1B | 61.20 | 1.20 | 100 | 1.60 | 98.0 | 0.0 | 100 | 3.2 |
| P, I, Q - (FR) | Qwen-3-32B | 31.08 | 72.80 | 99.60 | 89.20 | 98.40 | 0.0 | 98.0 | 84.80 |
| | Bloom-7.1B | 22.80 | 0.4 | 100 | 2.00 | 29.20 | 0.0 | 100 | 6 |
| P, I, Q - (JA) | Qwen-3-32B | 65.60 | 60.99 | 100 | 85.20 | 98.0 | 0.0 | 100 | 80.0 |
| | Bloom-7.1B | 18.00 | 0.40 | 100 | 0.80 | 81.20 | 0.0 | 100 | 1.6 |
| **Monolingual Average** | Qwen-3-32B | 65.56 | 66.60 | 99.47 | 90.53 | 65.87 | 0.13 | 99.20 | 86.80 |
| | Bloom-7.1B | 34.00 | 0.67 | 100 | 1.47 | 69.47 | 0.00 | 100.00 | 3.60 |
| P, I -(EN), Q(FR) | Qwen-3-32B | 7.60 | 59.60 | 99.60 | 87.20 | 7.60 | 0.0 | 97.2 | 86 |
| | Bloom-7.1B | 22.8 | 0.4 | 100 | 3.20 | 37.6 | 0.0 | 100 | 2.4 |
| P, I -(EN), Q(JA) | Qwen-3-32B | 6.08 | 54.40 | 90.40 | 86.80 | 100 | 0.0 | 100 | 83.20 |
| | Bloom-7.1B | 14.01 | 0.4 | 100 | 1.20 | 98.40 | 0.0 | 99.20 | 1.6 |
| **Code-Switched Average** | Qwen-3-32B | 6.80 | 57.00 | 95.00 | 87.00 | 53.80 | 0.00 | 98.60 | 84.60 |
| | Bloom-7.1B | 18.40 | 0.40 | 100 | 2.20 | 68.00 | 0.00 | 99.60 | 2.00 |

Table 9: Performance comparison of Qwen-3-32B and Bloom-7.1B on the XQuAD dataset in four languages (EN, ES, AR, HI). Qwen-3-32B exhibits high generalization and minimal performance degradation across SFT modes. In contrast, Bloom-7.1B struggles especially under zero-shot and random SFT, with F1 scores near zero in low-resource and distractor-heavy settings.

| Language | Model | Pre-Finetuning | | Full scope SFT | | Random Selective SFT | | Selective SFT | |
|---|---|---|---|---|---|---|---|---|---|
| | | Language Cons. (%) | F1 Score (%) | Language Cons. (%) | F1 Score (%) | Language Cons. (%) | F1 Score (%) | Language Cons. (%) | F1 Score (%) |
| P, I, Q, C - (EN) | Qwen-3-32B | 100 | 71.47 | 100 | 73.55 | 2.27 | 1.21 | 98.40 | 77.06 |
| | Bloom-7.1B | 99.42 | 11.67 | 100 | 25.08 | 61.76 | 0.0 | 100 | 24.06 |
| P, I, Q, C - (ES) | Qwen-3-32B | 41.68 | 56.17 | 100 | 69.27 | 22.61 | 0.45 | 100 | 62.09 |
| | Bloom-7.1B | 98.24 | 3.50 | 99.66 | 18.59 | 0.67 | 0.0 | 99.66 | 24.9 |
| P, I, Q, C - (AR) | Qwen-3-32B | 97.06 | 64.99 | 100 | 67.37 | 71.09 | 0.0 | 99.91 | 65.21 |
| | Bloom-7.1B | 97.23 | 0.31 | 100 | 11.92 | 72.86 | 0.0 | 100 | 16.58 |
| P, I, Q, C - (HI) | Qwen-3-32B | 85.46 | 29.51 | 100 | 18.22 | 95.80 | 0.0 | 100 | 17.09 |
| | Bloom-7.1B | 98.40 | 1.24 | 100 | 9.81 | 83.11 | 0.0 | 99.75 | 15.77 |
| **Monolingual Average** | Qwen-3-32B | 81.05 | 55.44 | 100.00 | 57.60 | 47.44 | 0.42 | 99.83 | 55.86 |
| | Bloom-7.1B | 98.32 | 4.18 | 99.91 | 16.85 | 54.10 | 0.00 | 99.85 | 20.83 |
| P, I - (EN), Q, C - (ES) | Qwen-3-32B | 1.60 | 65.42 | 100 | 68.23 | 97.06 | 2.53 | 100 | 66.19 |
| | Bloom-7.1B | 86.63 | 10.82 | 100 | 25.02 | 15.13 | 0.0 | 99.83 | 23.53 |
| P, I - (EN), Q, C - (AR) | Qwen-3-32B | 30.08 | 46.31 | 100 | 68.77 | 96.72 | 0.76 | 100 | 63.55 |
| | Bloom-7.1B | 58.57 | 3.42 | 100 | 19.60 | 17.31 | 0.0 | 99.91 | 19.01 |
| P, I - (EN), Q, C - (HI) | Qwen-3-32B | 1.34 | 46.23 | 100 | 18.60 | 100 | 0.0 | 100 | 30.85 |
| | Bloom-7.1B | 68.49 | 5.50 | 99.66 | 18.46 | 96.89 | 0.0 | 99.66 | 20.53 |
| **Code-Switched Average** | Qwen-3-32B | 11.01 | 52.65 | 100.00 | 51.87 | 97.93 | 1.10 | 100.00 | 53.53 |
| | Bloom-7.1B | 71.23 | 6.58 | 99.89 | 21.03 | 43.11 | 0.00 | 99.80 | 21.02 |
| P, I, Q - (ES), C - (EN) | Qwen-3-32B | 16.05 | 27.54 | 98.66 | 31.98 | 1.08 | 0.0 | 98.99 | 31.86 |
| | Bloom-7.1B | 70.08 | 1.37 | 97.56 | 12.58 | 0.67 | 0.0 | 98.99 | 9.24 |
| P, I, Q - (AR), C - (EN) | Qwen-3-32B | 62.27 | 9.81 | 30.84 | 13.25 | 20.67 | 0.0 | 97.73 | 14.88 |
| | Bloom-7.1B | 69.41 | 0.36 | 95.46 | 3.82 | 22.69 | 0.0 | 98.32 | 6.73 |
| P, I, Q - (HI), C - (EN) | Qwen-3-32B | 47.65 | 10.09 | 98.49 | 8.07 | 91.59 | 0.0 | 96.13 | 7.41 |
| | Bloom-7.1B | 68.82 | 0.28 | 98.06 | 4.47 | 55.13 | 0.0 | 97.39 | 4.87 |
| **English Distractor Average** | Qwen-3-32B | 41.99 | 15.81 | 75.99 | 17.77 | 37.78 | 0.00 | 97.62 | 18.05 |
| | Bloom-7.1B | 69.44 | 0.67 | 97.03 | 6.96 | 26.16 | 0.00 | 98.23 | 6.95 |

Table 10: Bloom-7.1B Layer-wise Ablation across `num_epochs` and `num_layers` to analyze the effect on language consistency (LC), task accuracy (TA). Models are fine-tuned on the MMLU business domain and evaluated on non-business domains under code-switched conditions. Varying the number of last layers (1–5) and training epochs (1–5) shows that fine-tuning just 3–5% of parameters yields high LC (>95%) with stable TA. The best combined scores emerge with 1–3 layers and 4–5 epochs, demonstrating that robust cross-domain, code-switched language control can be achieved with minimal parameter updates.

| Epochs | # of Last Layers | Lang Consistency (%) | | | | | | Task Accuracy (%) | | | | | | Avg-LC | Avg-TA | Combined |
|---|---|---|---|---|---|---|---|---|---|---|---|---|---|---|---|---|
| | | ES | FR | EN | HI | AR | JA | ES | FR | EN | HI | AR | JA | | | |
| Baseline → | | 48.08 | 26.92 | 99.04 | 34.62 | 41.35 | 32.69 | 25.96 | 27.88 | 22.12 | 18.27 | 7.69 | 27.88 | 47.12 | 21.63 | 34.37 |
| 1 | 1 | 99.04 | 99.04 | 85.58 | 100 | 100 | 99.04 | 23.08 | 23.08 | 13.46 | 12.50 | 3.85 | 12.50 | 97.12 | 14.74 | 55.93 |
| 1 | 2 | 99.04 | 100 | 74.04 | 100 | 100 | 100 | 25.00 | 22.12 | 10.58 | 12.50 | 0.96 | 17.31 | 95.51 | 14.74 | 55.13 |
| 1 | 3 | 100 | 99.04 | 40.38 | 99.04 | 100 | 98.08 | 20.19 | 21.15 | 11.54 | 12.50 | 9.62 | 12.50 | 89.42 | 14.58 | 52.00 |
| 1 | 4 | 98.08 | 99.04 | 73.08 | 99.04 | 100 | 98.08 | 18.27 | 13.46 | 9.62 | 12.50 | 3.85 | 16.35 | 94.55 | 12.34 | 53.45 |
| 1 | 5 | 100 | 99.04 | 82.69 | 100 | 100 | 98.08 | 17.31 | 22.12 | 15.38 | 16.35 | 6.73 | 12.50 | 96.63 | 15.06 | 55.85 |
| 2 | 1 | 99.04 | 99.04 | 79.81 | 99.04 | 98.08 | 99.04 | 16.35 | 28.85 | 17.31 | 13.46 | 8.65 | 12.50 | 95.67 | 16.19 | 55.93 |
| 2 | 2 | 100 | 100 | 67.31 | 99.04 | 100 | 100 | 24.04 | 27.88 | 13.46 | 8.65 | 7.69 | 16.35 | 94.39 | 16.35 | 55.37 |
| 2 | 3 | 98.08 | 98.08 | 69.23 | 99.04 | 99.04 | 98.08 | 31.73 | 16.35 | 12.50 | 16.35 | 5.77 | 14.42 | 93.59 | 16.19 | 54.89 |
| 2 | 4 | 98.08 | 99.04 | 34.62 | 99.04 | 99.04 | 97.12 | 27.88 | 23.08 | 6.73 | 24.04 | 13.46 | 19.23 | 87.82 | 19.07 | 53.45 |
| 2 | 5 | 99.04 | 99.04 | 60.58 | 100 | 100 | 97.12 | 26.92 | 20.19 | 15.38 | 18.27 | 12.50 | 14.42 | 92.63 | 17.95 | 55.29 |
| 3 | 1 | 99.04 | 100 | 84.62 | 98.08 | 100 | 97.12 | 25.96 | 23.08 | 12.50 | 12.50 | 7.69 | 12.50 | 96.47 | 15.71 | 56.09 |
| 3 | 2 | 98.08 | 97.12 | 69.23 | 99.04 | 100 | 97.12 | 28.85 | 25.00 | 14.42 | 18.27 | 14.42 | 14.42 | 93.43 | 19.23 | 56.33 |
| 3 | 3 | 99.04 | 97.12 | 58.65 | 100 | 100 | 98.08 | 26.92 | 24.04 | 12.50 | 19.23 | 13.46 | 15.38 | 92.15 | 18.59 | 55.37 |
| 3 | 4 | 99.04 | 99.04 | 51.92 | 99.04 | 100 | 100 | 24.04 | 25.96 | 10.58 | 15.38 | 14.42 | 16.35 | 91.51 | 17.79 | 54.65 |
| 3 | 5 | 100 | 100 | 62.50 | 100 | 99.04 | 98.08 | 30.77 | 20.19 | 10.58 | 20.19 | 18.27 | 17.31 | 93.27 | 19.55 | 56.41 |
| 4 | 1 | 98.08 | 97.12 | 86.54 | 99.04 | 100 | 98.08 | 25.00 | 20.19 | 11.54 | 15.38 | 8.65 | 13.46 | 96.47 | 15.71 | 56.09 |
| 4 | 2 | 99.04 | 100 | 64.42 | 100 | 100 | 100 | 30.77 | 23.08 | 18.27 | 25.00 | 16.35 | 15.38 | 93.59 | 21.47 | 57.53 |
| 4 | 3 | 97.12 | 98.08 | 58.65 | 98.08 | 99.04 | 99.04 | 28.85 | 30.77 | 9.62 | 17.31 | 17.31 | 11.54 | 91.67 | 19.23 | 55.45 |
| 4 | 4 | 100 | 99.04 | 50.00 | 100 | 99.04 | 99.04 | 24.04 | 26.92 | 15.38 | 17.31 | 16.35 | 14.42 | 91.19 | 19.07 | 55.13 |
| 4 | 5 | 99.04 | 99.04 | 47.12 | 98.08 | 99.04 | 97.12 | 28.85 | 28.85 | 11.54 | 21.15 | 12.50 | 21.15 | 89.90 | 20.67 | 55.29 |
| 5 | 1 | 98.08 | 98.08 | 87.50 | 100 | 100 | 100 | 26.92 | 23.08 | 15.38 | 20.19 | 17.31 | 18.27 | 96.79 | 20.19 | 58.49 |
| 5 | 2 | 98.08 | 99.04 | 75.96 | 98.08 | 99.04 | 98.08 | 32.69 | 20.19 | 15.38 | 18.27 | 13.46 | 13.46 | 94.71 | 18.91 | 56.81 |
| 5 | 3 | 100 | 97.12 | 73.08 | 99.04 | 99.04 | 100 | 29.81 | 24.04 | 13.46 | 15.38 | 20.19 | 21.15 | 94.71 | 20.67 | 57.69 |
| 5 | 4 | 99.04 | 99.04 | 71.15 | 100 | 97.12 | 100 | 23.08 | 29.81 | 13.46 | 16.35 | 22.12 | 18.27 | 94.39 | 20.51 | 57.45 |
| 5 | 5 | 99.04 | 100 | 53.85 | 99.04 | 99.04 | 99.04 | 21.15 | 25.96 | 10.58 | 21.15 | 21.15 | 19.23 | 91.67 | 19.87 | 55.77 |

Table 11: Layer-wise Selective SFT analysis of Qwen-3-32B on language consistency (LC) and task accuracy (TA) across six languages. The model is fine-tuned on the MMLU business domain and evaluated on non-business domains under code-switched conditions. Unlike the baseline, which shows strong TA (65.38%) but poor LC (24.52%), fine-tuning just 3–5% of the parameters quickly boosts LC to near-perfect levels (>99%) while preserving high TA. The best combined scores emerge with 2–3 layers and 4–5 epochs, indicating that minimal parameter updates are sufficient to reconcile Qwen's trade-off between task performance and language consistency.

| Epochs | # of Last Layers | Lang Consistency (%) | | | | | | Task Accuracy (%) | | | | | | Avg-LC | Avg-TA | Combined |
|---|---|---|---|---|---|---|---|---|---|---|---|---|---|---|---|---|
| | | ES | FR | EN | HI | AR | JA | ES | FR | EN | HI | AR | JA | | | |
| Baseline → | | 1.92 | 3.85 | 100 | 11.54 | 0.96 | 28.85 | 80.77 | 74.04 | 77.88 | 49.04 | 49.04 | 61.54 | 24.52 | 65.38 | 44.95 |
| 1 | 1 | 99.04 | 100 | 99.04 | 100 | 100 | 100 | 77.88 | 78.85 | 80.77 | 23.08 | 66.35 | 73.08 | 99.68 | 66.67 | 83.17 |
| 1 | 2 | 100 | 100 | 97.12 | 100 | 100 | 100 | 78.85 | 75.96 | 89.42 | 27.88 | 68.27 | 75.96 | 99.52 | 69.39 | 84.46 |
| 1 | 3 | 100 | 100 | 100 | 100 | 100 | 100 | 76.92 | 77.88 | 80.77 | 20.19 | 69.23 | 75.96 | 100 | 66.83 | 83.41 |
| 1 | 4 | 99.04 | 100 | 100 | 100 | 100 | 99.04 | 78.85 | 75.00 | 85.58 | 23.08 | 65.38 | 61.54 | 99.68 | 64.90 | 82.29 |
| 1 | 5 | 100 | 100 | 100 | 99.04 | 100 | 100 | 77.88 | 76.92 | 86.54 | 17.31 | 64.42 | 73.08 | 99.84 | 66.03 | 82.93 |
| 2 | 1 | 99.04 | 100 | 100 | 100 | 100 | 100 | 80.77 | 76.92 | 77.88 | 24.04 | 70.19 | 71.15 | 99.84 | 66.83 | 83.33 |
| 2 | 2 | 99.04 | 100 | 100 | 100 | 100 | 100 | 82.69 | 80.77 | 77.88 | 20.19 | 68.27 | 71.15 | 99.84 | 66.83 | 83.33 |
| 2 | 3 | 100 | 100 | 100 | 100 | 100 | 100 | 80.77 | 82.69 | 83.65 | 21.15 | 66.35 | 74.04 | 100 | 68.11 | 84.05 |
| 2 | 4 | 100 | 100 | 100 | 99.04 | 100 | 100 | 75.96 | 77.88 | 85.58 | 22.12 | 70.19 | 71.15 | 99.84 | 67.15 | 83.49 |
| 2 | 5 | 99.04 | 100 | 85.58 | 99.04 | 100 | 100 | 72.12 | 75.00 | 47.12 | 13.46 | 67.31 | 55.77 | 97.28 | 55.13 | 76.20 |
| 3 | 1 | 100 | 100 | 99.04 | 100 | 100 | 100 | 80.77 | 79.81 | 86.54 | 20.19 | 62.50 | 74.04 | 99.84 | 67.31 | 83.57 |
| 3 | 2 | 100 | 98.08 | 100 | 99.04 | 100 | 100 | 78.85 | 78.85 | 81.73 | 19.23 | 71.15 | 75.96 | 99.52 | 67.63 | 83.57 |
| 3 | 3 | 100 | 100 | 100 | 100 | 99.04 | 100 | 76.92 | 73.08 | 89.42 | 25.00 | 68.27 | 77.88 | 99.84 | 68.43 | 84.13 |
| 3 | 4 | 100 | 100 | 100 | 100 | 100 | 100 | 77.88 | 76.92 | 84.62 | 27.88 | 70.19 | 67.31 | 100 | 67.47 | 83.73 |
| 3 | 5 | 98.08 | 100 | 99.04 | 100 | 100 | 100 | 80.77 | 75.00 | 84.62 | 17.31 | 72.12 | 75.00 | 99.52 | 67.47 | 83.49 |
| 4 | 1 | 99.04 | 100 | 99.04 | 100 | 100 | 100 | 81.73 | 80.77 | 82.69 | 25.00 | 70.19 | 75.96 | 99.68 | 69.39 | 84.54 |
| 4 | 2 | 99.04 | 100 | 100 | 100 | 100 | 100 | 82.69 | 81.73 | 84.62 | 25.96 | 67.31 | 72.12 | 99.84 | 69.07 | 84.46 |
| 4 | 3 | 100 | 99.04 | 100 | 100 | 100 | 100 | 81.73 | 81.73 | 84.62 | 25.00 | 73.08 | 75.00 | 99.68 | 70.19 | 84.94 |
| 4 | 4 | 91.35 | 81.73 | 20.19 | 95.19 | 97.12 | 94.23 | 4.81 | 5.77 | 3.85 | 0.00 | 0.00 | 14.42 | 79.97 | 4.81 | 42.39 |
| 4 | 5 | 100 | 100 | 100 | 100 | 100 | 100 | 82.69 | 83.65 | 86.54 | 35.58 | 72.12 | 82.69 | 100 | 73.88 | 86.94 |
| 5 | 1 | 99.04 | 100 | 89.42 | 100 | 100 | 100 | 84.62 | 83.65 | 88.46 | 37.50 | 72.12 | 75.96 | 98.08 | 73.72 | 85.90 |
| 5 | 2 | 99.04 | 100 | 98.08 | 100 | 99.04 | 100 | 88.46 | 85.58 | 90.38 | 47.12 | 67.31 | 81.73 | 99.36 | 76.76 | 88.06 |
| 5 | 3 | 100 | 98.08 | 29.81 | 98.08 | 100 | 98.08 | 58.65 | 50.00 | 31.73 | 17.31 | 31.73 | 54.81 | 87.34 | 40.71 | 64.02 |
| 5 | 4 | 45.19 | 48.08 | 14.42 | 20.19 | 74.04 | 70.19 | 1.92 | 0.00 | 1.92 | 2.88 | 0.00 | 5.77 | 45.35 | 2.08 | 23.72 |
| 5 | 5 | 97.12 | 100 | 8.65 | 96.15 | 100 | 98.08 | 72.12 | 75.00 | 54.81 | 23.08 | 58.65 | 67.31 | 83.33 | 58.49 | 70.91 |

**Prompt: English Monolingual Direct Prompting**

You are given a multiple-choice question with four answer options: A, B, C, and D. Please choose the best answer based on your knowledge and reasoning ability.
**Response guidelines:**

- Your task is to carefully read the question and all answer choices, then determine which option best answers the question based on your knowledge and reasoning.

- Please consider the meaning of each choice and eliminate incorrect or less appropriate options using logical deduction or factual recall. If multiple answers seem plausible, select the one that is most accurate or comprehensive.

- Pay close attention to subtle distinctions in wording or concepts, as some questions may require domain-specific understanding or nuanced interpretation.

- After evaluating all options, select the single best answer and respond with only the corresponding letter: A, B, C, or D.

- Think step-by-step in the language of the question inside <thinking></thinking> and give your final answer in the language of the question inside <answer></answer>.

**Question:** {question}
**Answer choices:** {choices}
<thinking>

**Prompt: French Monolingual Direct Prompting**

Vous allez recevoir une question à choix multiples avec quatre options de réponse : A, B, C et D. Veuillez choisir la meilleure réponse en vous basant sur vos connaissances et votre capacité de raisonnement.
**Directives de réponse :**

- Votre tâche consiste à lire attentivement la question et toutes les options, puis à déterminer laquelle répond le mieux en fonction de vos connaissances et de votre raisonnement.

- Prenez en compte le sens de chaque option et éliminez celles qui sont incorrectes ou moins appropriées en utilisant la déduction logique ou des faits connus. Si plusieurs réponses semblent plausibles, choisissez celle qui est la plus précise ou la plus complète.

- Faites attention aux distinctions subtiles dans le libellé ou les concepts, car certaines questions peuvent nécessiter une compréhension spécialisée ou une interprétation nuancée.

- Après avoir évalué toutes les options, sélectionnez une seule réponse et répondez uniquement avec la lettre correspondante : A, B, C ou D.

- Réfléchis étape par étape dans la langue de la question à l'intérieur de <thinking></thinking> et donne ta réponse finale dans la langue de la question à l'intérieur de <answer></answer>.

**Question :** {question}
**Choix de réponses :** {choices}
<thinking>

---

**Prompt: Spanish Monolingual Direct Prompting**

Se te presenta una pregunta de opción múltiple con cuatro posibles respuestas: A, B, C y D. Por favor, elige la mejor respuesta basándote en tus conocimientos y capacidad de razonamiento.

**Instrucciones para la respuesta:**

- Tu tarea es leer cuidadosamente la pregunta y todas las opciones, y determinar cuál responde mejor basándote en tus conocimientos y razonamiento.

- Considera el significado de cada opción y elimina aquellas incorrectas o menos apropiadas utilizando la deducción lógica o el conocimiento factual. Si varias opciones parecen plausibles, selecciona la más precisa o completa.

- Presta especial atención a las diferencias sutiles en el lenguaje o los conceptos, ya que algunas preguntas pueden requerir comprensión específica del dominio o interpretación matizada.

- Después de evaluar todas las opciones, selecciona una sola respuesta y responde únicamente con la letra correspondiente: A, B, C o D.

- Piensa paso a paso en el idioma de la pregunta dentro de <thinking></thinking> y da tu respuesta final en el idioma de la pregunta dentro de <answer></answer>.

**Pregunta:** {question}
**Opciones de respuesta:** {choices}
<thinking>

---

**Prompt: Japanese Monolingual Direct Prompting**

多肢選択式の問題で、A、B、C、Dの4つの選択肢から回答してください。
あなたの知識と推論能力に基づき、最適な回答を選択してください。
回答ガイドライン：

- 質問とすべての選択肢をよく読み、あなたの知識と推論能力に基づき、どの選択肢が質問に最も適しているかを判断してください。

- 各選択肢の意味を考慮し、論理的推論または事実の想起を用いて、誤った選択肢や適切でない選択肢を除外してください。複数の回答が考えられる場合は、最も正確または包括的な選択肢を選択してください。

- 質問によっては、分野特有の理解や微妙な解釈が求められる場合があるので、言葉遣いや概念の微妙な違いにも注意してください。

- すべての選択肢を評価した後、最適な回答を1つ選び、対応する文字（A、B、C、またはD）のみで回答してください。

- <thinking></thinking> 内の質問の言語で段階的に考え、<answer></answer> 内の質問の言語で最終的な回答を記入してください。

質問: {question}
回答の選択肢: {choices}
<thinking>

**Prompt: Arabic Monolingual Direct Prompting**

يُعرَض عَلَيكَ سُؤَال اكهِتِيار مِن مُتَعَدّد بِأَربَعَت اكهِتِيَارات لِلاِجَابَ: أ، ب، ج، د.

يُرجَا اكهِتِيَار الاِجَابَ الانسَب بِنَاءَن عَلَا مَعرِفَتِكَ وَقُدرَتِكَ عَلَا التَّفكِير المَنطِقِي.

إِرشهَادَات الاِجَابَ:

مُهِمَّتُكَ هِيَ قِرَاءَت السُؤَال وَجَمِيع كهِيَارات الاِجَابَ بِعِنَايَ، تهُمَّ تَحدِيد الكهِيَار الانسَب بِنَاءَن عَلَا مَعرِفَتِكَ وَقُدرَتِكَ عَلَا التَّفكِير المَنطِقِي.

يُرجَا مُرَاعَات مَعنَا كُلّ كهِيَار وَاستِبعَاد الكهِيَارات غهَير السَحِيحَ او غهَير المُنَاسِب بِاستِكهدَام الاستِنتَاج المَنطِقِي او التَدهَكُّر. إِدهَا بَدَت الاِجَابَات المُتَعَدّد مَعقُولَ، فَكهتَر الاِجَابَ الاكتهَر دِقّ او سهُمُولَن.

إِنتَبِه جَيِّدَن لِلاكهِتِلَافَات الدَقِيقَ في التِيَاكهَة او المَفَاهِيم، فَقَد تَتَطَلَّب بَعد الاسئِلَ فَهمَن كهَاشَّن بِمَجَال مُعَيَّن او تَفسِيدَن دَقِيقَن.

بَعدَ تَقيِيم جَمِيع الكهِيَارات، اِكهتهَر الاِجَابَ الانسَب وَاجِب بِلحَرف المُقَابِل فَقَط: أ، ب، ج، أَو د.

فَكِّر كهَطوَ بِكهَطوَ بِللُّغَة السُؤَال ذَاكهِل

<thinking></thinking>,

وَقَدِّم اجَابَتَكَ النِهَائِّي بِللُّغَة السُؤَال ذَاكهِل

<answer></answer>.

أَلسُؤَال:

{question}

كهِيَارَات الاجَابَ:

{choices}
<thinking>

---

**Prompt: Hindi Monolingual Direct Prompting**

अअपअको एक बअहउवइकअलपइइयअ परअसहनअ दइयअअ गअयअअ हअइ जइसअकए चअअर उततअर वइकअलप हअइ म॰ॐ ा: भ: छ: अउरअ ध।करइपअयअअ अपअनए ज नअअनअ अउरअ तअरकअ कए अअदहअअर पअर सअबसए उपअयउकत उततअर चहउनए म।
उततअर दइसहअअनइरदएसहॐ

- सअबसए पअहअलए परअसहनअ अउरअ सअबहइइ उततअर वइकअलप दहयअअन सए पअ दहए म। पहइर सोचहए म कइ अअपअकअअ ज नअअनअ अउरअ तअरकअ कइस वइकअलप को सअबसए सअहइ बअनअअतए हअइ म।

- हअर वइकअलप कअअ मअतलअब सअमअजहए म अउरअ तअरकअ यअअ जअअनअकअअरइइ कअअ इसतएमअअल कअरअकए गअलअत यअअ कअम उपअयउकत वइकअलप हअ तअअ दएइन। अगअर कअइ वइकअलप सअहइ लअगए म: तो उनअमए म सए सअबसए सअ तइइक यअअ सअबसए पउउरअअ उततअर चहउनए म।

- सहअबदोन यअअ वइचहअअरोन कए चहहो थएस्चहहो थए पहअरकअ पअर दहयअअन दएइन: कयोनकइइ कउचहह सअवअअलोन मए म कहअअस जअअनअकअअरइइ यअअ नअअरुक वयअअकहयअअ चअहइयए हो सअकअतइइ हअइ।

- सअबहइइ वइकअलपोन पअर वइचहअअर कअरअनए कए बअअद सअबसए सअहइ वइकअलप चहउनए म अउरअ कएवअल सअमबअनद-हइत अक सअर लइकहए म॰ॐ ा: भ: छ: यअअ ध।

- <thinking></thinking> मए म उसइइ बहअअसहअअ मए म कअदअमस्दअरअस्कअदअम अपअनअअ तअरकअ लइकहए म अउरअ <answer></answer> मए म उसइइ बहअअसहअअ मए म अपअनअअ अअकहइरइइ उततअर दएइन।

परअसहनअॐ {question}
उततअर वइकअलपॐ {choices}
<thinking>

---

**Prompt: English-French Code-Switched Prompting**

You are given a multiple-choice question with four answer options: A, B, C, and D. Please choose the best answer based on your knowledge and reasoning ability.
Response guidelines:

- Your task is to carefully read the question and all answer choices, then determine which option best answers the question based on your knowledge and reasoning.

- Please consider the meaning of each choice and eliminate incorrect or less appropriate options using logical deduction or factual recall. If multiple answers seem plausible, select the one that is most accurate or comprehensive.

- Pay close attention to subtle distinctions in wording or concepts, as some questions may require domain-specific understanding or nuanced interpretation.

- After evaluating all options, select the single best answer and respond with only the corresponding letter: A, B, C, or D.

- Think step-by-step in the language of the question inside <thinking></thinking> and give your final answer in the language of the question inside <answer></answer>.

**Question:** Lequel des éléments suivants est la voie symplastique qui permet le déplacement du saccharose du site de photosynthèse des cellules du mésophylle vers le phloème ?
**Answer choices:**

- A. Les fibres, le parenchyme du phloème, la cellule compagne, le tube criblé

- B. Le parenchyme du phloème, les fibres, la gaine périvasculaire, les trachéides

- C. Les cellules compagnes, le parenchyme du phloème, les fibres, le tube criblé

- D. La gaine périvasculaire, le parenchyme du phloème, la cellule compagne, le tube criblé

<thinking>

**Prompt: English-Spanish Code-Switched Prompting**

You are given a multiple-choice question with four answer options: A, B, C, and D. Please choose the best answer based on your knowledge and reasoning ability.
Response guidelines:

- Your task is to carefully read the question and all answer choices, then determine which option best answers the question based on your knowledge and reasoning.

- Please consider the meaning of each choice and eliminate incorrect or less appropriate options using logical deduction or factual recall. If multiple answers seem plausible, select the one that is most accurate or comprehensive.

- Pay close attention to subtle distinctions in wording or concepts, as some questions may require domain-specific understanding or nuanced interpretation.

- After evaluating all options, select the single best answer and respond with only the corresponding letter: A, B, C, or D.

- Think step-by-step in the language of the question inside <thinking></thinking> and give your final answer in the language of the question inside <answer></answer>.

**Question:** ¿Cuál de las siguientes es la vía simplástica para el movimiento de la sacarosa desde el lugar de la fotosíntesis en las células del mesófilo hasta el floema?
**Answer choices:**

- A. Fibras, parénquima del floema, célula acompañante, tubo criboso.

- B. Parénquima del floema, fibras, haz vascular, traqueidas.

- C. células acompañantes, parénquima del floema, fibras, tubo criboso

- D. Haz vascular, parénquima del floema, célula acompañante, tubo criboso.

<thinking>

---

**Prompt: French Prompting with English-Distractor**

Lequel des éléments suivants est la voie symplastique qui permet le déplacement du saccharose du site de photosynthèse des cellules du mésophylle vers le phloème ?
**Choix de réponses :**

- A. Fibers, phloem parenchyma, companion cell, sieve tube

- B. Phloem parenchyma, fibers, bundle sheath, tracheids

- C. La gaine périvasculaire, le parenchyme du phloème, la cellule compagne, le tube criblé

- D. Companion cells, phloem parenchyma, fibers, sieve tube

Répondez en français. Réfléchissez étape par étape dans <thinking></thinking> et donnez votre réponse finale dans <answer></answer>.
<thinking>

---

**Prompt: Spanish Prompting with English-Distractor**

¿Cuál de las siguientes es la vía simplástica para el movimiento de la sacarosa desde el lugar de la fotosíntesis en las células del mesófilo hasta el floema?

**Opciones de respuesta:**

- A. Haz vascular, parénquima del floema, célula acompañante, tubo criboso.
- B. Fibers, phloem parenchyma, companion cell, sieve tube
- C. Phloem parenchyma, fibers, bundle sheath, tracheids
- D. Companion cells, phloem parenchyma, fibers, sieve tube

Responde en español. Piensa paso a paso dentro de <thinking></thinking> y da tu respuesta final en <answer></answer>.
<thinking>

---

**Prompt: French Prompting with English Bilingual Answer**

Lequel des éléments suivants est la voie symplastique qui permet le déplacement du saccharose du site de photosynthèse des cellules du mésophylle vers le phloème ?

**Choix de réponses :**

- A. Bundle sheath, phloem parenchyma, companion cell, sieve tube
- B. Le parenchyme du phloème, les fibres, la gaine périvasculaire, les trachéides
- C. Les cellules compagnes, le parenchyme du phloème, les fibres, le tube criblé
- D. La gaine périvasculaire, le parenchyme du phloème, la cellule compagne, le tube criblé

Répondez en français. Réfléchissez étape par étape dans <thinking></thinking> et donnez votre réponse finale dans <answer></answer>.
<thinking>

---

**Prompt: Spanish Prompting with English Bilingual Answer**

¿Cuál de las siguientes es la vía simplástica para el movimiento de la sacarosa desde el lugar de la fotosíntesis en las células del mesófilo hasta el floema?

**Opciones de respuesta:**

- A. Fibras, parénquima del floema, célula acompañante, tubo criboso.
- B. Parénquima del floema, fibras, haz vascular, traqueidas.
- C. Bundle sheath, phloem parenchyma, companion cell, sieve tube.
- D. Haz vascular, parénquima del floema, célula acompañante, tubo criboso.

Responde en español. Piensa paso a paso dentro de <thinking></thinking> y da tu respuesta final en <answer></answer>.
<thinking>

