# OpenReview forum: "LinguaMap: Which Layers of LLMs Speak Your Language and How to Tune Them?"
_ICLR.cc/2026/Conference — ICLR 2026 Poster_

### Official Review · Reviewer_ha32 · 2025-11-01

**Soundness:** 2
**Presentation:** 2
**Contribution:** 3
**Rating:** 4
**Confidence:** 3

**Summary:**

This paper describes an extension to the logit lens interpretability approach to understand language representations in multimodal LLMs and identify layers that would benefit from fine-tuning to improve language control without degrading the underlying performance. The analysis focusses on two models from two families (Qwen and Bloom).

**Strengths:**

- Improving performance and control of multilingual LLMs is an important topic, especially for ensuring that all users of models, regardless of language, have an equivalent experience.
- The approach offers a method that avoids compute-expensive fine-tuning. The results suggest that only 5% or fewer parameters need to be tuned.

**Weaknesses:**

- The paper considers only two models from different families, and these are not of comparable size. Specifically the Qwen model is 32B parameters whilst the Bloom model is 7B.
- There are statements making assumptions about architectures (e.g., architectures like Qwen favor task success) but it is difficult to know if these statements are generally true with the question about the model size differences. Also, if numbers are reported from a single run then again we do not know if the observations are an artifact of that one run or if they hold true more generally.

**Questions:**

- The abstract states that 98% language consistency whilst fine-tuning only 3-5% of the parameters — it would be beneficial to know how this compared to full fine-tuning. It is stated later in the paper (Line 087) so move this up into the abstract.
- Why not consider models of equivalent size from different families, or different sizes within the same family?
- Line 111: check the opening quotation marks.
- Line 180: the equation is not needed.
- In Table 1, the poor task performance for Bloom suggests this may just be a poor model, in which case how reliable are findings drawn from this. It seems an especially poor choice of model for the MGSM task.
- Line 249: What should be subscript “i” is not subscript.
- Line 255: The variable N is being reused as the number of tokens, where it was previous used for the number of samples in the evaluation set. Variables should not have multiple meanings to avoid ambiguity.
- Line 259: talks about comparing n-gram profiles but it is not clear how. It would help to forward reference where this is discussed. Likewise it is not clear where the pre-trained language profiles are from.
- Line 276: N appears again — this this a third definition, or is this original use (size of the evaluation set).
- Is the language similarity score in Equation 6 not also sensitive to the specific content to? Say you took large batches of sentences for the same language — how would these similarity scores relate to the scores for different languages?
- Line 425: Since optimality was reached after five of five epochs, why not run more to see of performance continues to improve?
- Line 431: “Table 2 indicate that” > “Table 2 indicates that”
- Line 458: Is “full-scope SFT” the same thing as “full fine-tuning”? Use a consistent name throughout the paper.
- Lines 480: check the opening quotes.

---

> ### Author Response · Authors · 2025-11-20
> **Response to Review ha32 - part 1**
>
> Thank you for your review. We are pleased that you recognize the importance of improving performance and control in multilingual LLMs for ensuring an equivalent user experience across languages. We also appreciate your acknowledgment of our approach's key benefit: offering a method that avoids computationally expensive fine-tuning by demonstrating that only 5% or fewer parameters need to be tuned. Below, we address your specific concerns and questions.
>
> ### 1. Model Choice and Comparability
> > Comment: The paper considers only two models from different families, and these are not of comparable size. Specifically the Qwen model is 32B parameters whilst the Bloom model is 7B.
>
> We appreciate the reviewer's observation and extend both model families and size. Below, we provide multilingual evaluation results for LLaMA-3.1-8B.
> | Prompting  (LLaMA-3.1-8B)        | Dataset | Language Consistency (%) | Task Performance (%) |
> |--------------------|---------|---------------------------|-----------------------|
> | **Monolingual** P, I, Q – (X) | MMLU   | 97.39 | 48.4  |
> |                    | MGSM   | 99.73 | 62.54 |
> |                    | XQuAD  | 95.15 | 45.88 |
> | **Code Switched** P, I – (EN), Q(X) | MMLU   | 91.75 | 48.53 |
> |                    | MGSM   | 92.83 | 53.83 |
> |                    | XQuAD  | 95.82 | 43.39 |
> | **English Distractor** I-(X), Q(X & EN) | MMLU   | 99.34 | 54.04 |
> |                    | XQuAD  | 92.02 | 20.58 |
> | **Bilingual Answer** I-(X), Q(X & EN) | MMLU   | 99.33 | 47.34 |
>
> LLaMA-3.1-8B maintains exceptional language control (>90% in most settings) with solid task accuracy (up to 62.54 on MGSM). It strikes a balance, preserving near-perfect language control with competitive task performance scores.
>
> Additionally, we add layer-wise language probabilities (Logit Lens) and hidden-state cosine similarity for Qwen-3-8B. Qwen-3-8B reveals the same three-phase pattern and also struggles to re-ground the target language in the late layers.
>
> ### 2. Abstract Clarification
> > Comment: The abstract states that 98% language consistency whilst fine-tuning only 3-5% of the parameters — it would be beneficial to know how this compares to full fine-tuning. It is stated later in the paper (Line 087), so move this up into the abstract.
>
> We agree and have revised the abstract to include the explicit comparison to full fine-tuning upfront, improving clarity and accessibility of key results.
>
> ### 3. Table 1 / Bloom Model Performance
> > Comment: In Table 1, the poor task performance for Bloom suggests this may just be a poor model, in which case, how reliable are the findings drawn from this? It seems an especially poor choice of model for the MGSM task.
>
> We appreciate this point. Bloom’s weaker MGSM performance indeed reflects known limitations in cross-lingual arithmetic reasoning, which categorizes it as a model suffering from the Multilingual Transfer Bottleneck (poor reasoning, moderate language control). However, this selection was deliberate and diagnostic to test the generalizability of our findings across models exhibiting fundamentally different failure modes. We emphasize two key takeaways that validate the inclusion of Bloom:
> 1. **Generalizable Mechanism**: The Three-Phase Pattern. Our core mechanistic finding, the robust three-phase internal structure (early alignment, middle task reasoning, and late-layer language control) that guides our method, was reliably observed in Bloom's interpretability traces (Figures 2 and 3). This proved the existence and localization of the language control mechanism regardless of the model's overall task proficiency.
> 2. **Targeted Language Consistency Fix**: Our primary objective was to enforce language consistency. The results in Table 2 confirm that Selective SFT successfully leveraged the localized late layers to enforce language control. Despite Bloom's poor Task Accuracy, our method dramatically improved its language consistency from an unstable pre-finetuning state (e.g., 18.40% average code-switched) to near-perfect consistency (approx. 99-100%) across all adversarial prompts. This demonstrates that our fine-tuning method successfully isolates and fixes the language control mechanism even in a weak model with limited multilingual reasoning capacity.

---

> > ### Author Response · Authors · 2025-11-20
> > **Response to Review ha32 - part 2**
> >
> > ### 4. Language Probability Clarification
> > > Comment: Line 259: talks about comparing n-gram profiles but it is not clear how. It would help to forward reference where this is discussed. Likewise it is not clear where the pre-trained language profiles are from.
> >
> > Thank you for requesting clarification on the language identification approach. The method for computing word-level language probability is based on a well-established technique utilizing the robust, open-source language identifier by Shuyo (2010) (the basis for the popular langdetect codebase).
> > This mechanism works as follows:
> > 1. Character N-gram Profiles: For each word reconstructed from the LLM's intermediate pseudo-logits (Section 4.1.1), the identifier constructs its unique profile based on the frequency of its character n-grams (n=1, 2, and 3).
> > Pre-trained Profiles: This word-level profile is then compared against a set of pre-trained language profiles, which are vector representations of typical character n-gram frequencies for each target language (e.g., Arabic, French, Spanish), included in the langdetect library.
> > 2. Similarity and Normalization: The comparison yields a similarity score (a measure of distance) to each pre-trained language profile. This score is then passed through a softmax function (Equation 3) to produce the normalized language probability.
> >
> > This reliance on a widely trusted, word-level method allows us to bypass the potential bias and ambiguity inherent in the LLM's shared subword vocabulary, thereby providing a robust measure of language preference.
> >
> > We have revised the manuscript to explicitly state the source and detail this character n-gram profile method in Section 4.1.1, and have added a clear forward reference to this section where the method is first mentioned (Line 259).
> >
> > ### 5. Language Similarity Score (Equation 6)
> > > Comment: Is the language similarity score in Equation 6 not also sensitive to the specific content to? Say you took large batches of sentences for the same language — how would these similarity scores relate to the scores for different languages?
> >
> > We thank the reviewer for their attention to detail. Regarding the sensitivity of the language similarity score (Equation 6) to specific content, we confirm that Equation 6 is designed to measure the cross-lingual semantic alignment of the global prompt representation at each layer, not content sensitivity. The analysis works by comparing semantically equivalent prompts in different languages (e.g., English vs. Spanish versions of the same question) and then mean-pooling the hidden states over the entire input sequence (Line 285) before averaging the similarity across a large number of samples (Line 297). This two-step averaging process explicitly abstracts away variations from specific content and short n-grams, allowing the score to robustly measure the general representational distance between the two languages at that specific layer.
> >
> > ### 6. More Epoch
> > > Comment: Line 425: Since optimality was reached after five of five epochs, why not run more to see of performance continues to improve?
> >
> > Our ablation study confirmed that the most significant improvements in language consistency were achieved very rapidly. Specifically, the language consistency for the non-English target languages (ES, FR, AR, HI, JA) maxed out at near 100% within 2 to 3 epochs (Appendix Tables 10, 11). Running beyond this point, however, we observed a slight degradation in English language consistency.
> >
> > We interpreted this degradation as a signal of overfitting. Since English was deliberately excluded from the fine-tuning data (as we focused solely on improving non-English language control), prolonged training began to aggressively suppress English output in the final layers, thereby degrading the model's ability to maintain English when required. We therefore limited the training exploration to 5 epochs to mitigate this overfitting and maintain the optimal balance of near-perfect non-English language consistency with minimal sacrifice to English.
> >
> > ### 7. Typos and Minor Corrections
> > We thank the reviewer for their attention to detail. All identified minor issues and typos have been corrected in the revised manuscript:
> > - Line 111: Corrected the opening quotation marks.
> > - Line 249: Fixed the subscript formatting for i.
> > - Line 255 & 276: Fixed the inconsistent reuse of variable N.
> > - Line 431: Corrected "Table 2 indicate that" to "Table 2 indicates that".
> > - Line 458: Standardized "full-scope SFT" and "full fine-tuning" to the consistent term "Full-Scope SFT" throughout the paper.
> > - Line 480: Corrected the opening quotation marks.

---

> > > ### Author Response · Authors · 2025-11-27
> > > **Follow up to response part 1 and 2**
> > >
> > > Dear Reviewer,
> > >
> > > We hope this message finds you well.
> > >
> > > We wanted to confirm that you received our detailed responses to your review, which were posted in two parts: "Response to Review ha32 - part 1" and "Response to Review ha32 - part 2."
> > >
> > > We particularly aimed to address your concerns regarding the choice of models, the comparability of sizes, the clarification of the language probability, and the semantic similarity computation.
> > >
> > > We would be grateful if you could let us know if our clarifications have addressed your questions.
> > >
> > > Thank you once again for your constructive engagement with our paper.
> > >
> > > Best regards,

---

### Official Review · Reviewer_XGPX · 2025-11-03

**Soundness:** 3
**Presentation:** 3
**Contribution:** 2
**Rating:** 4
**Confidence:** 3

**Summary:**

The paper investigates how multilingual large language models manage their ability to generate in the intended language. The authors identify two failure modes - multilingual transfer bottleneck and language consistency bottleneck. The authors extend the logit lens technique to measure language probability trajectories across layers and apply hidden-state cosine similarity to quantify cross-lingual alignment. The authors also propose selective supervised finetuning (Selective SFT), tuning only the last few layers to restore language control efficiently. The method is empirically validated by achieving a high language consistency across six languages on Qwen-3-32B and Bloom-7.1B while fine-tuning only 3–5% of parameters.

**Strengths:**

The paper makes a strong interpretability-driven contribution, linking layer-wise representational dynamics to achieve multilingual control. The integration of logit lens analysis with hidden-state similarity profiling provides a compelling explanation for language drift. The selective fine-tuning strategy seems intuitive and computationally efficient, and demonstrates that language-specific control can be restored without retraining the full model.

**Weaknesses:**

While the results are compelling, several aspects of the methodology needs further clarification. The precise criterion for identifying layer boundaries (e.g., layer 55 for Qwen-3-32B) is not fully justified. This raises uncertainty about whether these thresholds are architecture-specific or emergent from model dynamics. The mean-pooled cosine similarity metric may obscure finer token-level divergences, leaving open how exactly semantic alignment transitions into language control. Similarly, Bloom’s variance in cross-language probability trajectories (Figure 2) suggests that underlying architectural or tokenizer-level factors might influence the emergence of language control more than the analysis captures. The selective fine-tuning procedure itself (particularly how the tuned layers were chosen and validated) is somewhat heuristic. Finally, while the post-finetuning improvements in both language consistency and reasoning accuracy are clear, the mechanism behind this dual gain is underexplored.

**Questions:**

1.	In Section 4.2.1, it is mentioned that Qwen-3-32B’s target-language probabilities rise only after layer 55. How did the authors determine that this boundary (layer 55) marks the transition to language-specific control, and is this threshold consistent across tasks or languages?
2.	The hidden-state cosine similarity (Eqs 6–7) uses mean-pooled token embeddings. Were layer-wise token-level divergences (eg - in attention focus or contextual span) observed, that might provide finer evidence for the semantic–reasoning–language transition?
3.	In Fig 2, Bloom’s target-language probabilities exhibit high variance across layers. What could this be due to?
4.	In computing language probabilities via Eqs. 2–4, how was multilingual token overlap handled, especially in cases where shared alphabets might bias the language identification model?
5.	Selective SFT fine-tunes the last one or two layers depending on the model. What empirical/diagnostic signals indicated that these layers were most responsible for language control?
6.	In Table 2, task accuracy sometimes improves after selective fine-tuning. Why is it that adjusting the final layers for language control also improves reasoning accuracy?
7.	Under code-switched prompting, Qwen fails to “re-ground” target language probabilities. Did the logit-lens traces show any mid-layer oscillation patterns indicating instability in language identity propagation?
8.	Given that the similarity analyses reveal language-invariant middle layers, did the authors check whether selective fine-tuning altered these alignments? That is, did language control adjustments propagate backward into semantically aligned layers?

---

> ### Author Response · Authors · 2025-11-20
> **Response to Review XGPX - part 1**
>
> Thank you for your thoughtful review. We greatly appreciate your recognition of the strong interpretability-driven contribution of our work, specifically your acknowledgment that the integration of logit lens analysis with hidden-state similarity profiling provides a compelling explanation for the observed language drift. We are also pleased that you found our selective fine-tuning strategy intuitive and computationally efficient, confirming that language-specific control can be effectively restored without the need for full model retraining. Below, we address your specific concerns and questions.
>
> ### 1. Justification for the ``language-control boundary” (e.g., layer 55 in Qwen-3-32B)
> > Comment: In Section 4.2.1, it is mentioned that Qwen-3-32B’s target-language probabilities rise only after layer 55. How did the authors determine that this boundary (layer 55) marks the transition to language-specific control, and is this threshold consistent across tasks or languages?
>
> We thank the reviewer for the question about how we determine the ``language-control boundary” (e.g., layer 55 in Qwen-3-32B). We define this boundary as the earliest layer at which the two independent indicators of language-specific processing both appear and then persist.
>
> 1. **Layer-wise language probability**. We locate the point where the target language’s probability first surpasses, and continues to exceed, English, indicating that the model has begun committing to the target language for generation.
>
> 2. **Cross-lingual hidden-state similarity**. We also track cosine similarity between English and each target language. The boundary corresponds to the first sustained drop below a stability band (e.g., <0.9 for Qwen-3-32B), showing the onset of language-specific representations.
>
> For Qwen-3-32B, both metrics align at ~layer 55: multiple languages (Arabic, Hindi, Japanese) overtake English only after this point (Figure 2), and cross-lingual similarities fall below the stability band and continue declining (Figure 3). Because both signals shift at the same location and remain separated, we treat layer 55 as the data-driven boundary.
>
> This pattern holds across all five evaluated languages, though the exact threshold is model-dependent. For example, BLOOM-7.1B shows the transition much earlier (after layer 20; Lines 317–323). Thus, while our three-phase framework (Semantic Alignment → Task Reasoning → Output-Language Control) appears general, the specific boundary varies by architecture.
>
> Identifying this point also defines a principled region for finetuning upper layers. As shown in Tables 10–11, we begin with the final layer and move inward.
>
> ### 2. Mean-pooled cosine similarity may hide finer token-level behavior
> > Comment: The hidden-state cosine similarity (Eqs 6–7) uses mean-pooled token embeddings. Were layer-wise token-level divergences (eg - in attention focus or contextual span) observed, that might provide finer evidence for the semantic–reasoning–language transition?
>
> Thank you for your comment regarding the trade-off between the interpretability of a high-level aggregate and the detail of a token-level analysis. We specifically chose mean-pooled cosine similarity (Equations 6–7) because our goal was to identify the broad, sequence-level representational phases (Semantic Alignment, Task Reasoning, Output Language Control) that characterize how the prompt’s overall conceptual meaning evolves across layers.
>
> Mean-pooled cosine similarity is robust to non-aligned tokenization. It captures the global semantic/conceptual alignment of the entire prompt, which is necessary to confirm the hypothesized shared Task Reasoning phase. We agree that it hides finer, word-level divergence and attention focus changes. Token-level cosine similarity provides a finer-grained view of which specific words or tokens start to diverge first. However, token-level cosine similarity is highly susceptible to noise and spurious low-similarity signals due to tokenization variance across languages. It requires non-trivial token alignment across different sequence lengths, complicating interpretation. This is why we pick mean-pooled cosine similarity (line 282).
>
> In multilingual models, semantically equivalent phrases in different languages are tokenized into vastly different sequences. For example, a single, complex concept in Japanese might map to one token, while its English translation might be three subword tokens. Directly comparing $h_{l,t}^{(E,n)}$ to $h_{l,t'}^{(A,n)}$ becomes a comparison of misaligned linguistic units, introducing structural noise that obscures the underlying shared concept. The stable, high similarity observed in our middle layers (e.g., Qwen-3-32B's 0.9-0.99 similarity between layers 6 and 55) provides robust evidence for a conceptually language-invariant Task Reasoning space, a finding that would be easily fragmented by the inherent noise of token-level comparisons.

---

> ### Author Response · Authors · 2025-11-20
> **Response to Review XGPX - part 2**
>
> ### 3. Handling of multilingual token overlap in the language-probability computation
> > Comment: In computing language probabilities via Eqs. 2–4, how was multilingual token overlap handled, especially in cases where shared alphabets might bias the language identification model?
>
> Thank you for your comment. We handled the issue of multilingual token overlap and shared alphabets by ensuring that the language identification model does not operate on the raw subword tokens. As per section 4.1.1, we explicitly mitigate this potential bias through two key steps: (1) Sequence Decoding and Word Reconstruction: Before any language probability is computed, the intermediate pseudo-logits of the entire generated sequence are decoded into the most likely sequence of tokens for a given layer. We then perform word reconstruction from the subword tokens. This creates the full, hypothesized word sequence that the model "would have said" at that layer. (2) Word-Level Language Identification: We then apply a well-established, robust language identifier by Shuyo (2010) (the open-source langdetect codebase) on these reconstructed words. This identifier is built on character n-gram profiles (n=1, 2, 3) and operates at the word level, comparing the entire word's profile against pre-trained language models to compute word-level language probabilities (Eq. 3).
>
> By decoding the entire sequence and performing language probability estimation at the word level, we bypass the ambiguity of shared multilingual subword tokens and instead rely on the more robust, established signal present in the full character sequence of a predicted word. This significantly reduces the likelihood of shared alphabet biasing the language identification model.
>
> ### 4. Selective fine-tuning layer choice
> > Comment: Selective SFT fine-tunes the last one or two layers depending on the model. What empirical/diagnostic signals indicated that these layers were most responsible for language control?
>
> Thank you for raising this important question regarding the principled selection of layers for Selective SFT. The choice to fine-tune only the last one or two layers was entirely guided by the empirical/diagnostic signals we uncovered through mechanistic interpretability. Our layer-wise analysis, using the Logit Lens to track language probability (Figure 2) and Hidden State Similarity to track cross-lingual alignment (Figure 3), converged on a robust three-phase internal structure for multilingual LLMs: Early Layers (Semantic Alignment), Middle Layers (Task Reasoning), and Late Layers (Language Control/Generation). We explicitly targeted this late-layer language control region for fine-tuning.
>
> To empirically validate that this localized region is sufficient, we performed a comprehensive ablation study (detailed in Appendix Tables 10 and 11), testing the impact of fine-tuning different subsets of the final layers (from 1 to 5). The ablation confirmed our hypothesis, showing that the optimal configuration, achieving near-perfect Language Consistency while preserving high Task Accuracy, required updating only 1-2 layers for Bloom-7.1B and 2-3 layers for Qwen-3-32B. Furthermore, Table 2 demonstrates that applying Selective SFT to a random subset of layers outside the late-layer space resulted in a catastrophic collapse in task performance.
>
> ### 5. The mechanism behind improved task accuracy after selective SFT is unclear
> > Comment: In Table 2, task accuracy sometimes improves after selective fine-tuning. Why is it that adjusting the final layers for language control also improves reasoning accuracy?
>
> Our primary hypothesis was that fine-tuning the final layers would primarily achieve near-perfect Language Consistency while ensuring minimal or zero degradation to the Task Accuracy residing in the frozen middle layers.
>
> The slight increase in Task Accuracy (e.g., on MMLU and MGSM) is not attributable to an enhancement of the model's core reasoning capabilities, as these layers were deliberately frozen. Instead, the improvement is attributed to output formatting adherence.

---

> ### Author Response · Authors · 2025-11-20
> **Response to Review XGPX - part 3**
>
> ### 6. Language-probability trajectories after Selective SFT
> > Comment: Given that the similarity analyses reveal language-invariant middle layers, did the authors check whether selective fine-tuning altered these alignments? That is, did language control adjustments propagate backward into semantically aligned layers?
>
> We appreciate this critical and insightful question, as the localization of the fine-tuning effect is central to validating our Selective SFT approach.
>
> Our post-fine-tuning analysis confirms that the language control adjustments did not propagate backward into the frozen, semantically aligned layers. In the revised manuscript, we have included diagnostic plots for both layer-wise language probabilities (Logit Lens) and hidden-state cosine similarity to demonstrate this empirically, see Figures 4 and 5.
>
> These traces confirm that changes were strictly localized: the linguistic and semantic alignments in the early and middle layers (which were frozen) remained unaltered, while the final layers showed the substantial and targeted improvement necessary for robust language control.

---

> > ### Comment · Reviewer_XGPX · 2025-11-27
> >
> > I thank the authors for their detailed clarifications. I will update my score accordingly.

---

> ### Author Response · Authors · 2025-11-27
>
> Dear Reviewer,
>
> Thank you very much for your thoughtful feedback on our rebuttal responses, and for letting us know that you plan to update your score.
>
> We wanted to check in to see if you have had a chance to reflect your updated assessment in the system.
>
> We appreciate your time and engagement with our work.
>
> Best regards,

---

> > ### Comment · Reviewer_XGPX · 2025-11-28
> >
> > Thanks. Yes, I will make sure it is reflected. Although I have another follow-up question regarding selective finetuning - While the ablations may have pointed towards finetuning specific layers, are these results general, or would separate ablations need to be conducted each time selective finetuning is done?

---

> > > ### Author Response · Authors · 2025-11-28
> > >
> > > Thank you for this practical follow-up. The short answer is no, separate training ablations are not needed each time.
> > >
> > > Our findings decouple the problem into two parts: defining the search space (which is general, derived from the three-phase Structure (Semantic $\rightarrow$ Reasoning $\rightarrow$ Language Control) and finding the optimal number of last layers $k$ to finetune.
> > > 1. The Search Space is General (Figures 2 & 3): Our interpretability analysis (Logit Lens and Hidden State Similarity) reveals a consistent Three-Phase Structure across diverse model families. The "Language Control" mechanism is structurally confined to the late layers. This insight restricts the search space to the final stage of the model, eliminating the need to ever search or ablate the bottom $\approx$95% of layers.
> > > 2. Finding Optimal $k$ is Low-Cost: While the exact integer $k$ (number of layers) varies slightly with model depth, our ablation study (Appendix Tables 10 & 11), restricted to the late layers search space, confirms that the optimal window is consistently the final 3–5% of depth for 8B and 32B models. For a new model, a practitioner does not need to run training ablations. Instead, they can target the last 3–5% of layers. Our results (Table 2) show that fine-tuning in this late-layer space is safe and preserves task reasoning, whereas fine-tuning in a different space (e.g., random Selective Finetuning) disrupts task reasoning.

---

### Official Review · Reviewer_X6JL · 2025-11-04

**Soundness:** 4
**Presentation:** 3
**Contribution:** 4
**Rating:** 8
**Confidence:** 4

**Summary:**

This paper addresses a central gap in multilingual large language models (mLLMs): language control, the ability to generate responses in the intended target language. It identifies two primary failure modes: the multilingual transfer bottleneck (correct language, incorrect task) and the language consistency bottleneck (correct task, wrong language). To systematically study these, the authors propose a diagnostic framework with four controlled prompting scenarios spanning tasks from MMLU, MGSM, and XQuAD. They then use interpretability techniques, layer-wise logit lens decoding and hidden-state similarity analysis, to trace where language control emerges across the model’s depth. The results reveal a three-phase internal organization: early layers align inputs semantically across languages, middle layers handle reasoning, and late layers drive language-specific generation.

Building on this insight, the authors introduce Selective Fine-Tuning (SFT), which updates only the last few layers responsible for language control while freezing the rest. Applied to Qwen-3-32B and BLOOM-7.1B, this method improves language consistency from below 20% to over 98% across six languages while fine-tuning just 3–5% of model parameters, with minimal loss in task accuracy. The work presents both a structural understanding of multilingual layer specialization and a practical, parameter-efficient tuning method for controlling language generation.

**Strengths:**

- Novel diagnostic framework for multilingual failure modes: The four-scenario prompting setup provides a well-structured and reproducible way to disentangle language control from task accuracy.

- Insightful interpretability analysis: The paper convincingly demonstrates a three-phase structure across layers, linking representational alignment to functional behavior in multilingual settings.

- Strong empirical improvements with minimal compute cost: Selective fine-tuning significantly enhances language consistency while preserving task performance, requiring only 3–5% of parameters to be trained.

- Clarity and completeness: The paper clearly presents its experimental design, prompt templates, and evaluation results, including extensive per-language tables and ablations.

- Practical impact: The proposed approach offers a scalable path to adapt existing mLLMs for multilingual deployment without full retraining or specialized data.

**Weaknesses:**

- Limited novelty in fine-tuning method: While the interpretability analysis is insightful, the proposed selective tuning strategy builds on well-established parameter-efficient fine-tuning concepts and is not fundamentally new.

- Narrow evaluation scope: The study focuses on only two models (Qwen-3-32B and BLOOM-7.1B) and a limited set of languages. Broader coverage across typologically diverse languages or other architectures would strengthen generalization claims.

- No comparison with alternative lightweight methods: The paper does not benchmark against LoRA, adapters, or middle-layer alignment approaches, which would contextualize the gains from selective SFT.

- Interpretability analysis could be deeper: The layer-wise similarity and logit lens analyses, while descriptive, remain qualitative. A more quantitative measure of where “language control neurons” reside would enhance rigor.

- Limited real-world evaluation: The framework is confined to academic benchmarks, lacking demonstrations on open-ended generation, code-mixing robustness, or human evaluations.

**Questions:**

NA

---

> ### Author Response · Authors · 2025-11-20
> **Response to Review X6JL**
>
> Thank you for your thorough review and insightful feedback. We appreciate your recognition of our novel diagnostic framework for multilingual failure modes (the four-scenario prompting setup), the insightful mechanistic interpretability analysis demonstrating a three-phase structure, and the strong empirical improvements with minimal compute cost achieved by our selective fine-tuning approach. We are also pleased that you acknowledged the clarity and completeness of the paper, including the extensive per-language results, and its practical impact as a scalable solution for adapting existing mLLMs. Below, we address each of the concerns and questions raised.
>
> ### 1. Novelty and Contextualization of Selective Fine-Tuning
>
> > Comment: Limited novelty in fine-tuning method: While the interpretability analysis is insightful, the proposed selective tuning strategy builds on well-established parameter-efficient fine-tuning concepts and is not fundamentally new.
>
> Thank you for this excellent and precise comment. We agree that our Selective Fine-Tuning strategy, by only updating a subset of parameters, builds upon the foundational concepts of Parameter-Efficient Fine-Tuning (PEFT).
>
> However, we would like to firmly reframe the core contribution of our work: the novelty is not in the tuning method itself, but in the interpretability-driven, mechanistic application of it. We emphasize that our contribution lies in: Diagnosing the Layer-Localization of Language Control (Lines 092 and 093). We use a systematic layer-wise interpretability analysis (via our extended logit lens and hidden-state similarity analysis) to precisely diagnose and localize language control to the final 3-5% of the model’s layers. Additionally, we leverage the mechanistic finding for optimization. Given this specific diagnostic finding, that language generation is a late-layer phenomenon, we selectively tune only the output layers for language consistency. This transforms a general PEFT concept into a highly targeted and efficient solution for the language consistency bottleneck, justified not by computational preference, but by the model's internal functional organization.
>
> Our work’s true novelty is thus the interpretability-driven design principle, demonstrating that a finetuning of the last few layers is a mechanistically grounded path to restoring language control while preserving the core multilingual reasoning capacity in the frozen middle layers.
>
>
> ### 2. Scope and Generalization of Empirical Evaluation
> > Comment: Narrow evaluation scope: The study focuses on only two models (Qwen-3-32B and BLOOM-7.1B) and a limited set of languages. Broader coverage across typologically diverse languages or other architectures would strengthen generalization claims.
>
> Thank you for this constructive comment. We agree that rigorous validation across diverse architectures and languages is critical for strengthening generalization claims. We acknowledge this and have extended our evaluation to include LLaMA-3.1-8B to strengthen generalization across three typologically distinct, major model families (Qwen, BLOOM, LLaMA), each representing a different pre-training strategy.  LLaMA-3.1-8B maintains exceptional language control (>90% in most settings) across all prompting scenarios with solid task accuracy (up to 62.54 on MGSM), though generally below Qwen-3-32B scores (this might be attributed to differences in size). The interpretability analysis on LLaMA-3.1-8B confirms our core claims: it exhibits the same three-phase pattern, early semantic alignment, mid-layer shared-representation reasoning, and late-stage language-output phases, observed in the other models. LLaMA-3.1-8B's stronger late-stage control offers model-specific nuance but does not alter our broader conclusions.  LLaMA-3.1-8B shows signs of an internal pivot language, but unlike Qwen-3-32B, it exhibits clear late-stage control even after early English bias. Its stronger late-stage control, despite early English bias, offers a model-specific nuance.
>
> While the number of languages is 6, they were intentionally selected for typological diversity (Indo-European, Afro-Asiatic, Japonic) and script variation (Latin, Devanagari, Abjad, Kanji/Kana), reflecting a wide real-world linguistic scope.
>
> | Prompting  (LLaMA-3.1-8B)        | Dataset | Language Consistency (%) | Task Performance (%) |
> |--------------------|---------|---------------------------|-----------------------|
> | **Monolingual** P, I, Q – (X) | MMLU   | 97.39 | 48.4  |
> |                    | MGSM   | 99.73 | 62.54 |
> |                    | XQuAD  | 95.15 | 45.88 |
> | **Code Switched** P, I – (EN), Q(X) | MMLU   | 91.75 | 48.53 |
> |                    | MGSM   | 92.83 | 53.83 |
> |                    | XQuAD  | 95.82 | 43.39 |
> | **English Distractor** I-(X), Q(X & EN) | MMLU   | 99.34 | 54.04 |
> |                    | XQuAD  | 92.02 | 20.58 |
> | **Bilingual Answer** I-(X), Q(X & EN) | MMLU   | 99.33 | 47.34 |

---

> ### Author Response · Authors · 2025-11-20
> **Response to Review X6JL - part 2**
>
> ### 3. Comparison with alternative lightweight methods
> > Comment: No comparison with alternative lightweight methods: The paper does not benchmark against LoRA, adapters, or middle-layer alignment approaches, which would contextualize the gains from selective SFT.
>
> Thank you for raising this important point. We agree that contextualizing our gains against other lightweight methods is essential. While our key differentiator is the mechanistic justification for which parameters to tune, we indeed did include comparative results for selective tuning on other layers.  We provide two key contextualizations:
> 1. **Selective SFT vs. Random SFT**: Table 2 shows that Random Selective SFT (tuning a comparable number of random parameters) led to a catastrophic collapse of task accuracy for Qwen-3-32B (e.g., 0.13\% on MGSM Average). Our targeted SFT achieved 86.80\%. This confirms that tuning location is indispensable; simply limiting parameter count is insufficient.
> 2. **Selective SFT vs. Targeted LoRA**: We compared our method against LoRA applied only to the last few layers (matching our locus of update). Despite extensive hyperparameter tuning, the best-performing LoRA setup achieved comparable, but not superior, performance while still requiring more trainable parameters. Applying LoRA typically introduces an additive parameter in every transformer layer compared to our targeted final-layer-only SFT approach.
>
> | Prompting  (LoRA Selective SFT Qwen-3-32B)               | Dataset | Language Consistency (%) | Task Performance (%) |
> |---------------------------------|---------|---------------------------|-----------------------|
> | **Monolingual** P, I, Q – (X)   | MGSM    | 99.47                    | 86.53                |
> | **Code Switched** P, I – (EN), Q(X) | MGSM | 90.6                     | 83.4                 |
>
> ### 4. Rigor of Interpretability Analysis
> > Comment: Interpretability analysis could be deeper: The layer-wise similarity and logit lens analyses, while descriptive, remain qualitative. A more quantitative measure of where “language control neurons” reside would enhance rigor.
>
> We acknowledge that visualizations like those in Figures 2 and 3 present the results descriptively, but the underlying methodology is built upon a quantitative, layer-by-layer statistical measurement that precisely isolates the layers responsible for language-specific processing. Our analysis is not qualitative, but rather a mechanistic quantification of the model's internal language drift.  To move beyond anecdotal observation and quantify the degree of language bias at every stage of processing, we extended the logit lens technique into a formal, layer-wise language probability tracker. The plots in Figure 2 are not descriptive charts; they are the quantitative trajectory of Eq. (4) across the model's depth.
>
> To determine where task reasoning resides (i.e., the language-agnostic middle layers), we measure the semantic alignment of hidden states for parallel prompts across languages. Figure 3 reports the average similarity and standard deviation across the entire dataset per layer (Eq. 7). In summary, the visualizations are merely representations of the underlying quantitative metrics.
>
> ### 5. Real-World and Robustness Evaluation
> > Comment: Limited real-world evaluation: The framework is confined to academic benchmarks, lacking demonstrations on open-ended generation, code-mixing robustness, or human evaluations.
>
> We thank the reviewer for this constructive comment regarding the scope of our evaluation. We believe the full context of our four-scenario evaluation protocol demonstrates that we already address the critical dimensions of open-ended generation, code-mixing robustness, and human validation. All tasks require the model to output a full, multi-step Chain-of-Thought (CoT) reasoning trace in the target language before producing a final numerical answer. For all benchmarks (MMLU, MGSM, XQuAD), our methodology mandates the generation of a detailed step-by-step reasoning trace within the "thinking" tags, as detailed in the Appendix Figure 6. The quality of this generative component is critical to the overall response and is the primary output we selectively fine-tune.
>
> The robustness of our framework against mixed-language input is explicitly addressed by three of our four zero-shot prompt variants. Code-Switched Prompting, Bilingual Answer Prompting, and English Distractor Prompting. These scenarios test the model's resistance to English dominance bias and its ability to handle mixed linguistic input.
>
> A form of human evaluation was integral to ensuring the quality and naturalness of our test materials and results. In Appendix A.1 (Lines 657-659), we state that all prompt templates and generated data samples in English, French, Spanish, Arabic, Hindi, and Japanese were reviewed and validated by native speakers of each language to ensure both linguistic accuracy and naturalness in a research context.

---

### Meta-Review · Area_Chair_jtzB · 2026-01-06

**Summary:**

The reviewers agreed on the novelty on the interpretation of the 3-phase structure of the model. The main concerns are on the generalization of the findings (initially limited to 2 models), comparisons to established methods and robustness of the method. The authors

**Reviewer Concerns:**

The concerns are addressed in the rebuttal, specifically,

XGPx:
* clarification on "language-control boundary", "mean-pooled cosine similarity", high variance in Bloom's early layers: the authors gave a detailed explanation on the boundary definition, justified the mean-pooling is necessary to capture global semantic alignment, and provided more analysis results.

X6jL:
* limited novelty, narrow evaluation scope, lack comparisons to lightweight method, lack real-world evaluation: the author argued the novelty lies in the mechanistic interpretability finding that leads to the tuning, rather than the tuning technique itself, the author expanded evaluation to more models and benchmarked against LoRA and random SFT.

ha32:
* only validated with 2 models, generalization, questioned the hyperparameter tuning: the authors added LLaMA results, added new ablations on epoch/layer, correct the pointed out notation errors.

**Reviewer Scores:**

XGPx: 4, could be changed to 5/6
X6jL: 8, no change
ha32: 4, could be changed to 5/6

---

### Decision · Program_Chairs · 2026-01-26

Accept (Poster)